# On the Expressive Power of Geometric Graph Neural Networks

**Chaitanya K. Joshi**[*]
University of Cambridge, UK
chaitanya.joshi@cl.cam.ac.uk

**Cristian Bodnar**[*]
University of Cambridge, UK
cb2015@cam.ac.uk

**Simon V. Mathis**
University of Cambridge, UK
simon.mathis@cl.cam.ac.uk

**Taco Cohen**
Qualcomm AI Research, The Netherlands[†]
tacos@qti.qualcomm.com

**Pietro Liò**
University of Cambridge, UK
pietro.lio@cl.cam.ac.uk

## Abstract

The expressive power of Graph Neural Networks (GNNs) has been studied extensively through the lens of the Weisfeiler-Leman (WL) graph isomorphism test. Yet, many graphs in scientific and engineering applications come embedded in Euclidean space with an additional notion of geometric isomorphism, which is not covered by the WL framework. In this work, we propose a geometric version of the WL test (GWL) for discriminating geometric graphs while respecting the underlying physical symmetries: permutations, rotation, reflection, and translation. We use GWL to characterise the expressive power of GNNs that are invariant or equivariant to physical symmetries in terms of the classes of geometric graphs they can distinguish. This allows us to formalise the advantages of equivariant GNN layers over invariant ones: equivariant GNNs have greater expressive power as they enable propagating geometric information beyond local neighbourhoods, while invariant GNNs cannot distinguish graphs that are locally similar, highlighting their inability to compute global geometric quantities.

## 1 Introduction

Systems in biochemistry [1], material science [2], physical simulations [3], and multiagent robotics [4] contain both geometry and relational structure. Such systems can be modelled via *geometric graphs* embedded in Euclidean space. For example, molecules are represented as a set of nodes which contain information about each atom and its 3D spatial coordinates as well as other geometric quantities such as velocity or acceleration. Notably, the geometric attributes transform along with Euclidean transformations of the system, *i.e.* they are equivariant to symmetry groups of rotations, reflections, and translation. Standard Graph Neural Networks (GNNs) which do not take spatial symmetries into account are ill-suited for geometric graphs, as the geometric attributes would no longer retain their physical meaning and transformation behaviour [5, 6].

GNNs specialised for geometric graphs follow the message passing paradigm [7] where node features are updated in a permutation equivariant manner by aggregating features from local neighbourhoods. Crucially, in addition to permutations, the updated geometric features of the nodes retain the transformation behaviour of the initial attributes, *i.e.* they are also equivariant to the Lie group of rotations $(SO(d))$ or rotations and reflections $(O(d))$. We use $\mathfrak{G}$ as a generic symbol for these Lie groups. We consider two classes of GNNs for geometric graphs: (1) $\mathfrak{G}$-**equivariant models**, where the intermediate features and propagated messages are equivariant geometric quantities such as vectors or tensors [8–12]; and (2) $\mathfrak{G}$-**invariant models**, which only propagate local invariant scalar features such as distances and angles [13–15]. Both classes of architectures have shown promising empirical results for applications ranging from protein design [16, 17], molecular dynamics [18–20],

---

[*]Equal first authors.

[†]Qualcomm AI Research is an initiative of Qualcomm Technologies, Inc.

Joshi, Bodnar, et al., On the Expressive Power of Geometric Graph Neural Networks (Extended Abstract). Presented at the First Learning on Graphs Conference (LoG 2022), Virtual Event, December 9–12, 2022.

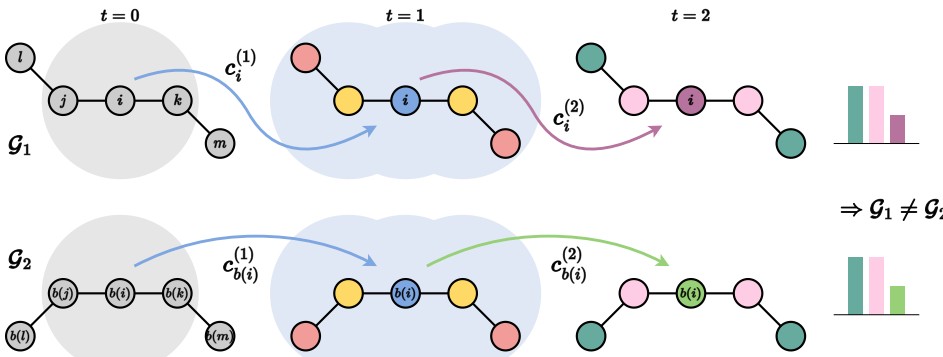

**Figure 1: Geometric Weisfeiler-Leman Test.** GWL distinguishes non-isomorphic geometric graphs $\mathcal{G}_1$ and $\mathcal{G}_2$ by injectively assigning colours to distinct neighbourhood patterns, up to global symmetries (here $\mathfrak{G} = O(d)$). Each iteration expands the neighbourhood from which geometric information can be gathered (shaded for node $i$). Example inspired by [18].

and electrocatalysis [21, 22]. At the same time, key theoretical questions remain unanswered: (1) How to characterise the *expressive power* of geometric GNNs? And (2) what is the tradeoff between $\mathfrak{G}$-equivariant and $\mathfrak{G}$-invariant GNNs?

The graph isomorphism problem [23] and the Weisfeiler-Leman (WL) [24] test for distinguishing non-isomorphic graphs have become a powerful tool for analysing the expressive power of non-geometric GNNs [25, 26]. The WL framework has been a major driver of progress in graph representation learning [27–31]. However, the WL framework does not directly apply to geometric graphs as they exhibit a stronger notion of isomorphism that also takes spatial symmetries into account.

**Contributions.** In this work, we study the expressive power of geometric GNNs from the perspective of discriminating non-isomorphic geometric graphs. We propose a geometric version of the Weisfeiler-Leman graph isomorphism test, termed GWL. We use GWL to formally characterise classes of graphs that can and cannot be distinguished by invariant and equivariant GNNs. We show how invariant GNNs have limited expressive power as they cannot distinguish graphs where one-hop local neighbourhoods are similar, while equivariant GNNs distinguish a larger class of graphs by propagating geometric vector quantities beyond local neighbourhoods.

For **Background and Preliminaries**, please see Appendix B.

## 2   The Geometric Weisfeiler-Leman Test

**Assumptions.** Analogous to the WL test, the geometric and scalar features the nodes are equipped with come from countable subsets $C \subset \mathbb{R}^d$ and $C' \subset \mathbb{R}$, respectively. As a consequence, when we require functions to be injective, we require them to be injective over the countable set of $\mathfrak{G}$-orbits that are obtained by acting with the symmetry group $\mathfrak{G}$ on the dataset. This mimics the practically relevant situation of finite datasets, in which we have a finite pool $\mathcal{P}$ of geometric graphs (and their symmetry transformations) which we would like to distinguish.

**Intuition.** For an intuition of how to generalise WL to geometric graphs, we note that WL uses a local, node-centric, procedure to update the colour of each node $i$ using the colours of its the 1-hop *neighbourhood* $\mathcal{N}_i$. In the geometric setting, $\mathcal{N}_i$ is an attributed point cloud around the central node $i$. As a result, each neighbourhood carries two types of information: (1) neighbourhood type (invariant to $\mathfrak{G}$) and (2) neighbourhood geometric orientation (equivariant to $\mathfrak{G}$). From an axiomatic point of view, our generalisation of the WL aggregation procedure must meet two properties:

**Property 1: Orbit injectivity of colours.** If two neighbourhoods are the same up to an action of $\mathfrak{G}$ (*e.g.* rotation), then the colours of the corresponding central nodes should be the same. Thus, the colouring must be $\mathfrak{G}$-orbit injective – which also makes it $\mathfrak{G}$-invariant – over the countable set of all orbits of neighbourhoods in our dataset.

**Property 2: Preservation of local geometry.** A key property of WL is that the aggregation is injective. A $\mathfrak{G}$-invariant colouring procedure that purely satisfies Property 1 is not sufficient because, by definition, it loses spatial properties of each neighbourhood such as the relative pose or orientation [32]. Thus, we must additionally update auxiliary *geometric information* variables in a way that is $\mathfrak{G}$-equivariant and injective.

**Geometric Weisfeiler-Leman (GWL).** These intuitions motivate the following definition of the GWL test. At initialisation, we assign to each node $i \in \mathcal{V}$ a scalar node colour $c_i \in C'$ and an auxiliary object $\boldsymbol{g}_i$ containing the geometric information associated to it:

$$c_i^{(0)} := \text{HASH}(\boldsymbol{s}_i), \quad \boldsymbol{g}_i^{(0)} := \left( c_i^{(0)}, \vec{\boldsymbol{v}}_i \right), \tag{1}$$

where HASH denotes an injective map over the scalar attributes $\boldsymbol{s}_i$ of node $i$. To define the inductive step, assume we have the colours of the nodes and the associated geometric objects at iteration $t-1$. Then, we can aggregate the geometric information around node $i$ into a new object as follows:

$$\boldsymbol{g}_i^{(t)} := \left( (c_i^{(t-1)}, \boldsymbol{g}_i^{(t-1)}) , \ \{\!\{ (c_j^{(t-1)}, \boldsymbol{g}_j^{(t-1)}, \vec{\boldsymbol{x}}_{ij}) \mid j \in \mathcal{N}_i \}\!\} \right), \tag{2}$$

Here $\{\!\{\cdot\}\!\}$ denotes a multiset – a set in which elements may occur more than once. Importantly, the group $\mathfrak{G}$ can act on the geometric objects above inductively by acting on the geometric information inside it. This amounts to rotating (or reflecting) the entire $t$-hop neighbourhood contained inside:

$$\mathfrak{g} \cdot \boldsymbol{g}_i^{(0)} := \left( c_i^{(0)}, \ \boldsymbol{Q}_{\mathfrak{g}} \vec{\boldsymbol{v}}_i \right), \quad \mathfrak{g} \cdot \boldsymbol{g}_i^{(t)} := \left( (c_i^{(t-1)}, \mathfrak{g} \cdot \boldsymbol{g}_i^{(t-1)}), \ \{\!\{ (c_j^{(t-1)}, \mathfrak{g} \cdot \boldsymbol{g}_j^{(t-1)}, \boldsymbol{Q}_{\mathfrak{g}} \vec{\boldsymbol{x}}_{ij}) \mid j \in \mathcal{N}_i \}\!\} \right)$$

Clearly, the aggregation building $\boldsymbol{g}_i$ for any time-step $t$ is injective and $\mathfrak{G}$-equivariant. Finally, we can compute the node colours at iteration $t$ for all $i \in \mathcal{V}$ by aggregating the geometric information in the neighbourhood around node $i$:

$$c_i^{(t)} := \text{I-HASH}^{(t)} \left( \boldsymbol{g}_i^{(t)} \right), \tag{3}$$

by using a $\mathfrak{G}$-orbit injective and $\mathfrak{G}$-invariant function that we denote by I-HASH. That is for any geometric objects $\boldsymbol{g}, \boldsymbol{g}'$, I-HASH$(\boldsymbol{g}) = $ I-HASH$(\boldsymbol{g}')$ if and only if there exists $\mathfrak{g} \in \mathfrak{G}$ such that $\boldsymbol{g} = \mathfrak{g} \cdot \boldsymbol{g}'$. Note that I-HASH is an idealised $\mathfrak{G}$-orbit injective function, similar to the HASH function used in WL, which is not necessarily continuous.

**Overview.** With each iteration, $\boldsymbol{g}_i^{(t)}$ aggregates geometric information in progressively larger $t$-hop subgraph neighbourhoods $\mathcal{N}_i^{(t)}$ around the node $i$. The node colours summarise the structure of these $t$-hops via the $\mathfrak{G}$-invariant aggregation performed by I-HASH. The procedure terminates when the partitions of the nodes induced by the colours do not change from the previous iteration. Finally, given two geometric graphs $\mathcal{G}$ and $\mathcal{H}$, if there exists some iteration $t$ for which $\{\!\{ c_i^{(t)} \mid i \in \mathcal{V}(\mathcal{G}) \}\!\} \neq \{\!\{ c_i^{(t)} \mid i \in \mathcal{V}(\mathcal{H}) \}\!\}$, then GWL deems the two graphs as being geometrically non-isomorphic. Otherwise, we say the test cannot distinguish the two graphs.

**Invariant GWL.** Since we are interested in understanding the role of $\mathfrak{G}$-equivariance, we also consider a more restrictive Invariant GWL (IGWL) that only updates node colours using the $\mathfrak{G}$-orbit injective I-HASH function and does not propagate geometric information:

$$c_i^{(t)} := \text{I-HASH} \left( (c_i^{(t-1)}, \vec{\boldsymbol{v}}_i) , \ \{\!\{ (c_j^{(t-1)}, \vec{\boldsymbol{v}}_j, \vec{\boldsymbol{x}}_{ij}) \mid j \in \mathcal{N}_i \}\!\} \right). \tag{4}$$

**IGWL with $k$-body scalars.** In order to further analyse the construction of the node colouring function I-HASH, we consider IGWL$_{(k)}$ based on the maximum number of nodes involved in the computation of $\mathfrak{G}$-invariant scalars (also known as the 'body order' [33]):

$$c_i^{(t)} := \text{I-HASH}_{(k)} \left( (c_i^{(t-1)}, \vec{\boldsymbol{v}}_i) , \ \{\!\{ (c_j^{(t-1)}, \vec{\boldsymbol{v}}_j, \vec{\boldsymbol{x}}_{ij}) \mid j \in \mathcal{N}_i \}\!\} \right), \tag{5}$$

and I-HASH$_{(k+1)}$ is defined as:

$$\text{HASH} \left( \{\!\{ \text{I-HASH} \left( (c_i^{(t-1)}, \vec{\boldsymbol{v}}_i), \{\!\{ (c_{j_1}^{(t-1)}, \vec{\boldsymbol{v}}_{j_1}, \vec{\boldsymbol{x}}_{ij_1}), \ldots, (c_{j_k}^{(t-1)}, \vec{\boldsymbol{v}}_{j_k}, \vec{\boldsymbol{x}}_{ij_k}) \}\!\} \right) \mid \boldsymbol{j} \in (\mathcal{N}_i)^k \}\!\} \right),$$

where $\boldsymbol{j} = [j_1, \ldots, j_k]$ are all possible $k$-tuples formed of elements of $\mathcal{N}_i$. Therefore, IGWL$_{(k)}$ is now constrained to extract information only from all the possible $k$-sized tuples of nodes (including the central node) in a neighbourhood. For instance, I-HASH$_{(2)}$ can identify neighbourhoods only up to pairwise distances among the central node and any of its neighbours (*i.e.* a 2-body scalar), while I-HASH$_{(3)}$ up to distances and angles formed by any two edges (*i.e.* a 3-body scalar). Notably, distances and angles alone are incomplete descriptors of local geometry [34, 35]. Therefore, I-HASH$_{(k)}$ with lower $k$ makes the colouring weaker.

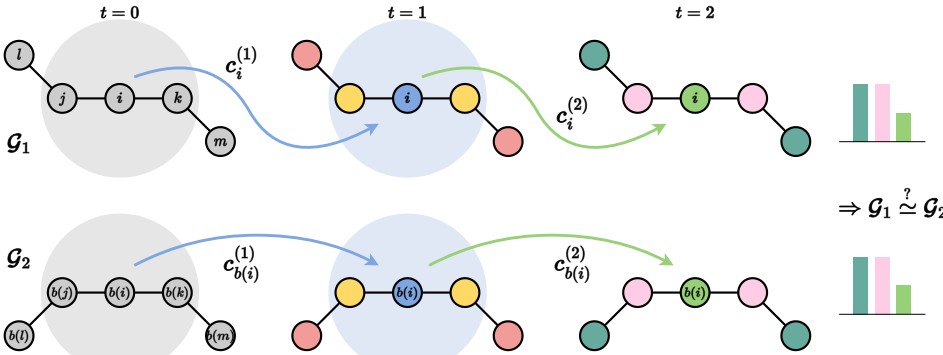

**Figure 2: Invariant GWL Test.** IGWL cannot distinguish $\mathcal{G}_1$ and $\mathcal{G}_2$ as they are 1-hop identical: The $\mathfrak{G}$-orbit of the 1-hop neighbourhood around each node is the same, and IGWL cannot propagate geometric orientation information beyond 1-hop (here $\mathfrak{G} = O(d)$).

## 2.1 What Geometric Graphs can GWL and IGWL Distinguish?

In order to formalise the expressive power of GWL and IGWL, let us consider what geometric graphs can and cannot be distinguished by the tests. As a simple first observation, we note that when all coordinates and vectors are set equal to zero GWL coincides with the standard WL. In this *edge case*, GWL has the same expressive power as WL.

Next, let us consider consider the simplified setting of two geometric graphs $\mathcal{G}_1 = (\boldsymbol{A}_1, \boldsymbol{S}_1, \vec{\boldsymbol{V}}_1, \vec{\boldsymbol{X}}_1)$ and $\mathcal{G}_2 = (\boldsymbol{A}_2, \boldsymbol{S}_2, \vec{\boldsymbol{V}}_2, \vec{\boldsymbol{X}}_2)$ such that the underlying attributed graphs $(\boldsymbol{A}_1, \boldsymbol{S}_1)$ and $(\boldsymbol{A}_2, \boldsymbol{S}_2)$ are isomorphic. This case frequently occurs in chemistry, where molecules occur in different conformations, but with the same graph topology given by the covalent bonding structure. Recall that each iteration of GWL aggregates geometric information $\boldsymbol{g}_i^{(k)}$ from progressively larger neighbourhoods $\mathcal{N}_i^{(k)}$ around the node $i$, and distinguishes (sub-)graphs via comparing $\mathfrak{G}$-orbit injective colouring of $\boldsymbol{g}_i^{(k)}$. We say $\mathcal{G}_1$ and $\mathcal{G}_2$ are *$k$-hop distinct* if for all graph isomorphisms $b$, there is some node $i \in \mathcal{V}_1, b(i) \in \mathcal{V}_2$ such that the corresponding $k$-hop subgraphs $\mathcal{N}_i^{(k)}$ and $\mathcal{N}_{b(i)}^{(k)}$ are distinct. Otherwise, we say $\mathcal{G}_1$ and $\mathcal{G}_2$ are *$k$-hop identical* if all $\mathcal{N}_i^{(k)}$ and $\mathcal{N}_{b(i)}^{(k)}$ are identical up to group actions.

We can now formalise what geometric graphs can and cannot be distinguished by GWL. Proofs are available in Appendix D.1.

**Proposition 1.** *GWL can distinguish any $k$-hop distinct geometric graphs $\mathcal{G}_1$ and $\mathcal{G}_2$ where the underlying attributed graphs are isomorphic, and $k$ iterations are sufficient.*

**Proposition 2.** *Up to $k$ iterations, GWL cannot distinguish any $k$-hop identical geometric graphs $\mathcal{G}_1$ and $\mathcal{G}_2$ where the underlying attributed graphs are isomorphic.*

Additionally, we can state the following results about the more constrained IGWL.

**Proposition 3.** *IGWL can distinguish any 1-hop distinct geometric graphs $\mathcal{G}_1$ and $\mathcal{G}_2$ where the underlying attributed graphs are isomorphic, and 1 iteration is sufficient.*

**Proposition 4.** *Any number of iterations of IGWL cannot distinguish any 1-hop identical geometric graphs $\mathcal{G}_1$ and $\mathcal{G}_2$ where the underlying attributed graphs are isomorphic.*

An example illustrating Propositions 1 and 4 is shown in Figures 1 and 2, respectively.

We can now consider the more general case where the underlying attributed graphs for $\mathcal{G}_1 = (\boldsymbol{A}_1, \boldsymbol{S}_1, \vec{\boldsymbol{V}}_1, \vec{\boldsymbol{X}}_1)$ and $\mathcal{G}_2 = (\boldsymbol{A}_2, \boldsymbol{S}_2, \vec{\boldsymbol{V}}_2, \vec{\boldsymbol{X}}_2)$ are non-isomorphic and constructed from point clouds using radial cutoffs, as conventional for biochemistry and material science applications.

**Proposition 5.** *Assuming geometric graphs are constructed from point clouds using radial cutoffs, GWL can distinguish any geometric graphs $\mathcal{G}_1$ and $\mathcal{G}_2$ where the underlying attributed graphs are non-isomorphic. At most $k_{Max}$ iterations are sufficient, where $k_{Max}$ is the maximum graph diameter among $\mathcal{G}_1$ and $\mathcal{G}_2$.*

These results enable us to compare the expressive powers of GWL and IGWL.

**Theorem 6.** *GWL is strictly more powerful than IGWL.*

This statement formalises the advantage of 𝔊-equivariant intermediate layers for graphs and geometric data, as prescribed in the Geometric Deep Learning blueprint [6], in addition to echoing similar intuitions in the computer vision community. As remarked by Hinton et al. [32], translation invariant models do not understand the relationship between the various parts of an image (colloquially called the "Picasso problem"). Similarly, our results point to IGWL failing to understand how the various 1-hops of a graph are stitched together.

Finally, we identify a setting where this distinction between the two approaches disappears.

**Proposition 7.** *IGWL has the same expressive power as GWL for fully connected geometric graphs.*

## 2.2 Characterising the Expressive Power of Geometric GNNs

We would like to characterise the maximum expressive power of geometric GNNs based on the GWL test. Firstly, we show that any message passing 𝔊-equivariant GNN can be at most as powerful as GWL in distinguishing non-isomorphic geometric graphs. Proofs are available in Appendix D.2.

**Theorem 8.** *Any pair of geometric graphs distinguishable by a 𝔊-equivariant GNN is also distinguishable by GWL.*

With a sufficient number of iterations, the output of 𝔊-equivariant GNNs can be equivalent to GWL if certain conditions are met regarding the aggregate, update and readout functions.

**Proposition 9.** *𝔊-equivariant GNNs have the same expressive power as GWL if the following conditions hold: (1) The aggregation $\mathrm{AGG}$ is an injective, 𝔊-equivariant multiset function. (2) The scalar part of the update $\mathrm{UPD}_s$ is a 𝔊-orbit injective, 𝔊-invariant multiset function. (3) The vector part of the update $\mathrm{UPD}_v$ is an injective, 𝔊-equivariant multiset function. (4) The graph-level readout $\mathfrak{f}$ is an injective multiset function.*

Similar statements can be made for 𝔊-invariant GNNs and IGWL. Thus, we can directly transfer our results about GWL and IGWL to the class of GNNs bounded by the respective tests.

This has several interesting practical implications, discussed in Appendix C. Most notably, Propositions 1, 4, and 10 highlight a critical theoretical limitation of 𝔊-invariant GNNs: their inability to compute global and non-local geometric quantities. Our results suggest that 𝔊-equivariant GNNs should be preferred when working with large geometric graphs such as macromolecules with thousands of nodes, where message passing is restricted to local radial neighbourhoods around each node. Motivated by these limitations, two straightforward approaches to improving 𝔊-invariant GNNs may be: (1) pre-computing non-local geometric properties as input features; and (2) working with fully connected geometric graphs, as Proposition 7 suggests.

## 3 Discussion

We propose the Geometric Weisfeiler-Leman test (GWL), a novel theoretical tool to characterise the expressive power of geometric GNNs from the perspective of geometric graph isomorphism. We believe our work fills a key research gap as geometric graphs come with both relational structure and geometric attributes, making theoretical tools for standard GNNs inapplicable in this increasingly relevant real-world setting. We use GWL to provide practical insights into the workings and theoretical limitations of geometric GNNs. Most notably, we formalise the advantages of equivariant GNNs over invariant GNNs: invariant models have limited expressive power as they only reason locally via scalar quantities, while equivariant GNNs can distinguish a larger class of graphs by propagating geometric vector quantities beyond local neighbourhoods.

**Future Work.** GWL provides an abstraction to study the theoretical limits of geometric GNNs. In practice it is challenging to build provably powerful GNNs that satisfy the conditions of Proposition 9 as GWL relies on perfect colouring and aggregation functions to identify distinct neighbourhoods and propogate their geometric orientation information, respectively. Based on the intuitions gained from GWL, future work will explore building maximally powerful, *practical* geometric GNNs for applications in biochemistry, material science, and multiagent robotics, and better characterise the trade-offs related to practical implementation choices.

## Acknowledgements

We would like to thank Andrew Blake, Challenger Mishra, Charles Harris, Dávid Kovács, Erik Thiede, Gabor Csanyi, Hannes Stärk, Ilyes Batatia, Iulia Duta, Justin Tan, Mario Geiger, Petar Veličković, Ramon Vinãs, Rob Hesselink, Soledad Villar, Weihua Hu, and the anonymous reviewers for helpful comments and discussions. CKJ was supported by the A*STAR Singapore National Science Scholarship (PhD). SVM was supported by the UKRI Centre for Doctoral Training in Application of Artificial Intelligence to the study of Environmental Risks (EP/S022961/1).

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

## A   Related Work

Literature on the *completeness* of atom-centred interatomic potentials has focused on distinguishing 1-hop local neighbourhoods (point clouds) around atoms by building spanning sets for continuous, 𝔊-equivariant multiset functions [35–38]. Recent theoretical work on geometric GNNs and their universality has shown that architectures such as TFN, GemNet and GVP-GNN [10, 21, 39, 40] can be universal approximators of continuous, 𝔊-equivariant or 𝔊-invariant multiset functions over point clouds, *i.e.* fully connected graphs. In contrast, the GWL framework studies the expressive power of geometric GNNs operating on sparse graphs from the perspective of discriminating geometric graphs. The discrimination lens is potentially more granular and practically insightful than universality. A model may either be universal or not. On the other hand, there could be multiple degrees of discrimination depending on the classes of geometric graphs that can and cannot be distinguished, which our work aims to formalise.

Geometric graph matching has also been studied from the perspective of finding global isometries in the computer vision community [41]. Our notion of geometric graph isomorphism is more general as it considers local message passing procedures as well as both scalar and geometric node attributes.

# B  Background and Preliminaries

## B.1  Graph Isomorphism and the Weisfeiler-Leman Test

An attributed graph $\mathcal{G} = (\boldsymbol{A}, \boldsymbol{S})$ is a set $\mathcal{V}$ of $n$ nodes connected by edges. $\boldsymbol{A}$ denotes an $n \times n$ adjacency matrix where each entry $a_{ij} \in \{0, 1\}$ indicates the presence or absence of an edge connecting nodes $i$ and $j$. The matrix of *scalar* features $\boldsymbol{S} \in \mathbb{R}^{n \times f}$ stores attributes $\boldsymbol{s}_i \in \mathbb{R}^f$ associated with each node $i$, where the subscripts index rows and columns in the corresponding matrices. The graph isomorphism problem asks whether two graphs are the same, but drawn differently [23]. Two attributed graphs $\mathcal{G}, \mathcal{H}$ are *isomorphic* (denoted $\mathcal{G} \simeq \mathcal{H}$) if there exists an edge-preserving bijection $b : \mathcal{V}(\mathcal{G}) \to \mathcal{V}(\mathcal{H})$ such that $\boldsymbol{s}_i^{(\mathcal{G})} = \boldsymbol{s}_{b(i)}^{(\mathcal{H})}$.

The *Weisfeiler-Leman test* (WL) is an algorithm for testing whether two (attributed) graphs are isomorphic [24]. At iteration zero the algorithm assigns a *colour* $c_i^{(0)} \in C$ from a countable space of colours $C$ to each node $i$. Nodes are coloured the same if their features are the same, otherwise, they are coloured differently. In subsequent iterations $t$, WL iteratively updates the node colouring by producing a new $\boldsymbol{c}_i^{(t)} \in C$:

$$\boldsymbol{c}_i^{(t)} := \text{HASH}\left(\boldsymbol{c}_i^{(t-1)}, \{\!\!\{\boldsymbol{c}_j^{(t-1)} \mid j \in \mathcal{N}_i\}\!\!\}\right), \tag{6}$$

where HASH is an injective map (*i.e.* a perfect hash map) that assigns a unique colour to each input. The test terminates when the partition of the nodes induced by the colours becomes stable. Given two graphs $\mathcal{G}$ and $\mathcal{H}$, if there exists some iteration $t$ for which $\{\!\!\{c_i^{(t)} \mid i \in \mathcal{V}(\mathcal{G})\}\!\!\} \neq \{\!\!\{c_i^{(t)} \mid i \in \mathcal{V}(\mathcal{H})\}\!\!\}$, then the graphs are not isomorphic. Otherwise, the WL test is inconclusive, and we say it cannot distinguish the two graphs.

The graph isomorphism problem and WL have become a powerful tool for characterising the theoretical limits of GNNs [42, 43]. It was shown by Xu et al. [25], Morris et al. [26] that message passing GNNs are at most as powerful as WL at distinguishing non-isomorphic graphs, *i.e.* the expressive power of GNNs is upper-bounded by WL. GNNs can have the same expressive power as WL if their aggregate, update, and readout are injective functions over multisets. The WL framework has since become a major driver of progress in designing more expressive GNNs [27–31].

## B.2  Group Theory

We assume basic familiarity with group theory, see Zee [44] for an overview. We denote the action of a group $\mathfrak{G}$ on a space $X$ by $\mathfrak{g} \cdot x$. If $\mathfrak{G}$ acts on spaces $X$ and $Y$, we say: (1) A function $f : X \to Y$ is $\mathfrak{G}$-*invariant* if $f(\mathfrak{g} \cdot x) = f(x)$, *i.e.* the output remains unchanged under transformations of the input; and (2) A function $f : X \to Y$ is $\mathfrak{G}$-*equivariant* if $f(\mathfrak{g} \cdot x) = \mathfrak{g} \cdot f(x)$, *i.e.* a transformation of the input must result in the output transforming equivalently.

We also consider $\mathfrak{G}$-*orbit injective* functions. The $\mathfrak{G}$-*orbit* of $x \in X$ is $\mathcal{O}_{\mathfrak{G}}(x) = \{\mathfrak{g} \cdot x \mid \mathfrak{g} \in \mathfrak{G}\} \subseteq X$. When $x$ and $x'$ are part of the same orbit, we write $x \simeq x'$. We say a function $f : X \to Y$ is $\mathfrak{G}$-*orbit injective* if we have $f(x_1) = f(x_2)$ if and only if $x_1 \simeq x_2$ for any $x_1, x_2 \in X$. Necessarily, such a function is $\mathfrak{G}$-invariant, since $f(\mathfrak{g} \cdot x) = f(x)$.

We work with the permutation group over $n$ elements $S_n$ and the Lie groups $\mathfrak{G} = SO(d)$ or $\mathfrak{G} = O(d)$. Invariance to the translation group $T(d)$ is conventionally handled using relative positions or by subtracting the centre of mass from all nodes positions. Given one of the standard groups above, for an element $\mathfrak{g}$ we denote by $\boldsymbol{M}_{\mathfrak{g}}$ (or another capital letter) its standard matrix representation.

## B.3  Geometric Graphs

A geometric graph $\mathcal{G} = (\boldsymbol{A}, \boldsymbol{S}, \vec{\boldsymbol{V}}, \vec{\boldsymbol{X}})$ is an attributed graph that is also decorated with geometric attributes: node coordinates $\vec{\boldsymbol{X}} \in \mathbb{R}^{n \times d}$ and (optionally) vector features $\vec{\boldsymbol{V}} \in \mathbb{R}^{n \times d}$ (*e.g.* velocity, acceleration). Without loss of generality, we work with a single vector feature per node. Our setup generalises to multiple vector features or higher-order tensors. In biochemistry and material science, the conventional procedure for constructing the geometric graph $\mathcal{G} = (\boldsymbol{A}, \boldsymbol{S}, \vec{\boldsymbol{V}}, \vec{\boldsymbol{X}})$ is via the underlying point cloud $(\boldsymbol{S}, \vec{\boldsymbol{V}}, \vec{\boldsymbol{X}})$ using a predetermined radial cutoff $r$. Thus, the adjacency matrix is defined as $a_{ij} = 1$ if $\|\vec{\boldsymbol{x}}_i - \vec{\boldsymbol{x}}_j\|_2 \leq r$, or 0 otherwise, for all $a_{ij} \in \boldsymbol{A}$.

The geometric attributes transform as follows under the action of the relevant groups.

- The permutation group $S_n$ acts via a permutation matrix $\boldsymbol{P}_\sigma$ on the graph attributes as $\boldsymbol{P}_\sigma \mathcal{G} := (\boldsymbol{P}_\sigma \boldsymbol{A} \boldsymbol{P}_\sigma^\top, \boldsymbol{P}_\sigma \boldsymbol{S}, \boldsymbol{P}_\sigma \vec{\boldsymbol{V}}, \boldsymbol{P}_\sigma \vec{\boldsymbol{X}})$;

- The group of rotation $SO(d)$ or rotations and reflections $O(d)$, denoted interchangeably by $\mathfrak{G}$, acts via an orthogonal transformation matrix $\boldsymbol{Q}_{\mathfrak{g}} \in \mathfrak{G}$ on the vector feature $\vec{\boldsymbol{V}}$ as $\vec{\boldsymbol{V}} \boldsymbol{Q}_{\mathfrak{g}}$, and on the coordinates $\vec{\boldsymbol{X}}$ as $\vec{\boldsymbol{X}} \boldsymbol{Q}_{\mathfrak{g}}$;

- The group of translations $T(d)$ acts via a translation vector $\vec{\boldsymbol{t}} \in T(d)$ on the coordinates $\vec{\boldsymbol{X}}$ as $\vec{\boldsymbol{x}}_i + \vec{\boldsymbol{t}}$ for all nodes $i$.

Two geometric graphs $\mathcal{G}$ and $\mathcal{H}$ are *geometrically isomorphic* if there exists an attributed graph isomorphism $b$ such that the geometric attributes are equivalent, up to global group actions $\boldsymbol{Q}_{\mathfrak{g}} \in \mathfrak{G}$ and $\vec{\boldsymbol{t}} \in T(d)$:

$$\left( \boldsymbol{s}_i^{(\mathcal{G})}, \vec{\boldsymbol{v}}_i^{(\mathcal{G})}, \vec{\boldsymbol{x}}_i^{(\mathcal{G})} \right) = \left( \boldsymbol{s}_{b(i)}^{(\mathcal{H})}, \boldsymbol{Q}_{\mathfrak{g}} \vec{\boldsymbol{v}}_{b(i)}^{(\mathcal{H})}, \boldsymbol{Q}_{\mathfrak{g}} (\vec{\boldsymbol{x}}_{b(i)}^{(\mathcal{H})} + \vec{\boldsymbol{t}}) \right) \quad \text{for all } i \in \mathcal{V}(\mathcal{G}). \tag{7}$$

Geometric graph isomorphism and distinguishing (sub-)graph geometries has important practical implications for representation learning. For *e.g.*, in molecular systems, an ideal architecture should map distinct local structural environments around atoms to distinct embeddings in representation space [34, 35].

## B.4  Geometric Graph Neural Networks

GNNs specialised for geometric graphs can be categorised according to the type of intermediate representations: (1) *Equivariant models*, where the intermediate features and propagated messages are geometric quantities; and (2) *Invariant models*, which only propagate local invariant scalar features such as distances and angles.

**$\mathfrak{G}$-invariant GNNs.**  $\mathfrak{G}$-invariant GNN layers aggregate scalar quantities from local neighbourhoods via scalarising the geometric information. Scalar features are update from iteration $t$ to $t + 1$ via learnable aggregate and update functions, AGG and UPD, respectively:

$$\boldsymbol{s}_i^{(t+1)} := \text{UPD} \left( \boldsymbol{s}_i^{(t)}, \text{AGG} \left( \{\!\{ (\boldsymbol{s}_i^{(t)}, \boldsymbol{s}_j^{(t)}, \vec{\boldsymbol{v}}_i, \vec{\boldsymbol{v}}_j, \vec{\boldsymbol{x}}_{ij}) \mid j \in \mathcal{N}_i \}\!\} \right) \right). \tag{8}$$

For *e.g.*, SchNet [13] uses relative distances $\|\vec{\boldsymbol{x}}_{ij}\|$ to scalarise local geometric information:

$$\boldsymbol{s}_i^{(t+1)} := \boldsymbol{s}_i^{(t)} + \sum_{j \in \mathcal{N}_i} f_1 \left( \boldsymbol{s}_j^{(t)}, \|\vec{\boldsymbol{x}}_{ij}\| \right) \qquad \text{(SchNet)} \tag{9}$$

DimeNet [15] uses both distances and angles $\vec{\boldsymbol{x}}_{ij} \cdot \vec{\boldsymbol{x}}_{ik}$ among triplets, as follows:

$$\boldsymbol{s}_i^{(t+1)} := \sum_{j \in \mathcal{N}_i} f_1 \left( \boldsymbol{s}_i^{(t)}, \boldsymbol{s}_j^{(t)}, \sum_{k \in \mathcal{N}_i \setminus \{j\}} f_2 \left( \boldsymbol{s}_j^{(t)}, \boldsymbol{s}_k^{(t)}, \|\vec{\boldsymbol{x}}_{ij}\|, \vec{\boldsymbol{x}}_{ij} \cdot \vec{\boldsymbol{x}}_{ik} \right) \right) \quad \text{(DimeNet)} \tag{10}$$

The updated scalar features are both $\mathfrak{G}$-invariant and $T(d)$-invariant as the only geometric information used are the relative distances and angles, both of which remain unchanged under the action of $\mathfrak{G}$ or translations. For both $\mathfrak{G}$-invariant and $\mathfrak{G}$-equivariant architectures (described subsequently), the scalar features $\{\boldsymbol{s}_i^{(T)}\}$ at the final iteration $T$ are mapped to graph-level features via a permutation-invariant readout $f : \mathbb{R}^{n \times f} \to \mathbb{R}^{f'}$.

**$\mathfrak{G}$-equivariant GNNs using cartesian vectors.**  $\mathfrak{G}$-equivariant GNN layers update both scalar and vector features by propagating scalar as well as vector messages, $\boldsymbol{m}_i^{(t)}$ and $\vec{\boldsymbol{m}}_i^{(t)}$, respectively:

$$\boldsymbol{m}_i^{(t)}, \vec{\boldsymbol{m}}_i^{(t)} := \text{AGG} \left( \{\!\{ (\boldsymbol{s}_i^{(t)}, \boldsymbol{s}_j^{(t)}, \vec{\boldsymbol{v}}_i^{(t)}, \vec{\boldsymbol{v}}_j^{(t)}, \vec{\boldsymbol{x}}_{ij}) \mid j \in \mathcal{N}_i \}\!\} \right) \qquad \text{(Aggregate)} \tag{11}$$

$$\boldsymbol{s}_i^{(t+1)}, \vec{\boldsymbol{v}}_i^{(t+1)} := \text{UPD} \left( (\boldsymbol{s}_i^{(t)}, \vec{\boldsymbol{v}}_i^{(t)}), (\boldsymbol{m}_i^{(t)}, \vec{\boldsymbol{m}}_i^{(t)}) \right) \qquad \text{(Update)} \tag{12}$$

For *e.g.*, PaiNN [18] interaction layers aggregate scalar and vector features via learnt filters conditioned on the relative distance:

$$\boldsymbol{m}_i^{(t)} := \boldsymbol{s}_i^{(t)} + \sum_{j \in \mathcal{N}_i} f_1\left(\boldsymbol{s}_j^{(t)}, \|\vec{\boldsymbol{x}}_{ij}\|\right) \tag{13}$$

$$\vec{\boldsymbol{m}}_i^{(t)} := \vec{\boldsymbol{v}}_i^{(t)} + \sum_{j \in \mathcal{N}_i} f_2\left(\boldsymbol{s}_j^{(t)}, \|\vec{\boldsymbol{x}}_{ij}\|\right) \odot \vec{\boldsymbol{v}}_j^{(t)} + \sum_{j \in \mathcal{N}_i} f_3\left(\boldsymbol{s}_j^{(t)}, \|\vec{\boldsymbol{x}}_{ij}\|\right) \odot \vec{\boldsymbol{x}}_{ij} \tag{14}$$

E-GNN [11] and GVP-GNN [10] use similar operations. The update step applies a gated non-linearity [45] on the vector features, which learns to scale their magnitude using their norm concatenated with the scalar features:

$$\boldsymbol{s}_i^{(t+1)} := \boldsymbol{m}_i^{(t)} + f_4\left(\boldsymbol{m}_i^{(t)}, \|\vec{\boldsymbol{m}}_i^{(t)}\|\right), \qquad \vec{\boldsymbol{v}}_i^{(t+1)} := \vec{\boldsymbol{m}}_i^{(t)} + f_5\left(\boldsymbol{m}_i^{(t)}, \|\vec{\boldsymbol{m}}_i^{(t)}\|\right) \odot \vec{\boldsymbol{m}}_i^{(t)}. \tag{15}$$

The updated scalar features are both $\mathfrak{G}$-invariant and $T(d)$-invariant as the only geometric information used is the relative distances, while the updated vector features are $\mathfrak{G}$-equivariant and $T(d)$-invariant as they aggregate $\mathfrak{G}$-equivariant, $T(d)$-invariant vector quantities from the neighbours.

**$\mathfrak{G}$-equivariant GNNs using spherical tensors.** Another example of $\mathfrak{G}$-equivariant GNNs is the e3nn framework [46], which can be used to instantiate Tensor Field Network [8], Cormorant [9], SEGNN [12], and MACE [20]. These models use higher order spherical tensors $\tilde{\boldsymbol{h}}_{i,l} \in \mathbb{R}^{2l+1 \times f}$ as node feature, starting from order $l = 0$ up to arbitrary $l = L$. The first two orders correspond to scalar features $\boldsymbol{s}_i$ and vector features $\vec{\boldsymbol{v}}_i$, respectively. The higher order tensors $\tilde{\boldsymbol{h}}_i$ are updated via tensor products of neighbourhood features $\tilde{\boldsymbol{h}}_j$ for all $j \in \mathcal{N}_i$ with the higher order spherical harmonic representations $Y$ of the relative displacement $\frac{\vec{\boldsymbol{x}}_{ij}}{\|\vec{\boldsymbol{x}}_{ij}\|} = \hat{\boldsymbol{x}}_{ij}$:

$$\tilde{\boldsymbol{h}}_i^{(t+1)} := \tilde{\boldsymbol{h}}_i^{(t)} + \sum_{j \in \mathcal{N}_i} Y\left(\hat{\boldsymbol{x}}_{ij}\right) \otimes_{\boldsymbol{w}} \tilde{\boldsymbol{h}}_j^{(t)}, \tag{16}$$

where the weights $\boldsymbol{w}$ of the tensor product are computed via a learnt radial basis function of the relative distance, *i.e.* $\boldsymbol{w} = f\left(\|\vec{\boldsymbol{x}}_{ij}\|\right)$. To obtain the entry $m_3 \in \{-l_3, \dots, +l_3\}$ for the order-$l_3$ part of the updated higher order tensors $\tilde{\boldsymbol{h}}_i^{(t+1)}$, we can expand the tensor product in equation 16 as:

$$\tilde{\boldsymbol{h}}_{i,l_3 m_3}^{(t+1)} := \tilde{\boldsymbol{h}}_{i,l_3 m_3}^{(t)} + \sum_{l_1 m_1, l_2 m_2}^{l_3 m_3} C_{l_1 m_1, l_2 m_2}^{l_3 m_3} \sum_{j \in \mathcal{N}_i} f_{l_1 l_2 l_3}\left(\|\vec{\boldsymbol{x}}_{ij}\|\right) Y_{l_1}^{m_1}\left(\hat{\boldsymbol{x}}_{ij}\right) \tilde{\boldsymbol{h}}_{j,l_2 m_2}^{(t)}, \tag{17}$$

where $C_{l_1 m_1, l_2 m_2}^{l_3 m_3}$ are the Clebsch-Gordan coefficients ensuring that the updated features are equivariant. Notably, when restricting the tensor product to only scalars (up to $l = 0$), we obtain updates of the form similar to equation 9. Similarly, when using only scalars and vectors (up to $l = 1$), we obtain updates of the form similar to equation 13, equation 14 and equation 15.

## C Understanding the Design Space of Geometric GNNs via GWL

**Overview.** We can use the GWL framework to better understand key design choices for building geometric GNNs [33]: (1) Depth or number of layers; and (2) Body order of invariant scalars. In doing so, we formalise theoretical limitations of current architectures and provide practical implications. Proofs are available in Appendix D.3.

### C.1 Role of Depth: Propagating Geometric Information

Each iteration of GWL expands the neighbourhood from which geoemtric information can be gathered. We leveraged this construction in Section 2.1 to formalise the number of GWL iterations required to distinguish classes of geometric graphs.

Consequently, stacking multiple $\mathfrak{G}$-equivariant GNN layers enables the computation of compositional geometric features. This can be understood via a geometric version of computation trees [47], as illustrated in Figure 3. A computation tree $\mathcal{T}_i^{(t)}$ represents the maximum information contained in GWL/IGWL colours or GNN features for node $i$ at iteration $t$ by an 'unrolling' of the message passing procedure. GWL, IGWL, and the corresponding classes of GNNs can be intuitively understood as

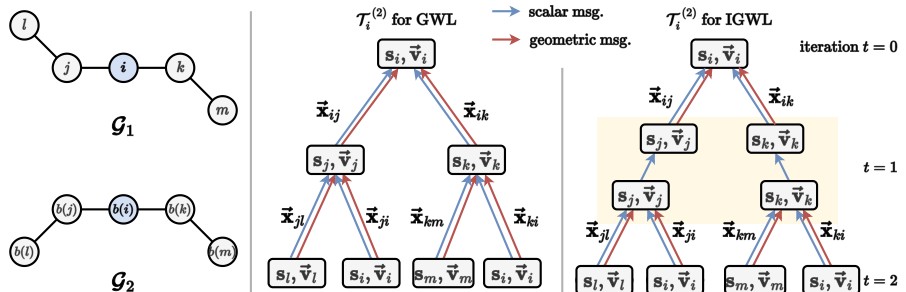

**Figure 3: Geometric Computation Trees for GWL and IGWL.** Unlike GWL, geometric orientation information cannot flow from the leaves to the root in IGWL, restricting its expressive power. IGWL cannot distinguish $\mathcal{G}_1$ and $\mathcal{G}_2$ as all 1-hop neighbourhoods are computationally identical.

colouring geometric computation trees. Critically, geometric orientation information cannot flow from one level to another in the computation trees for IGWL and $\mathfrak{G}$-invariant GNNs, as they only update scalar information.

As a result, even the most powerful $\mathfrak{G}$-invariant GNNs are restricted in their ability to compute global and non-local geometric properties.

**Proposition 10.** *IGWL and $\mathfrak{G}$-invariant GNNs cannot decide several geometric graph properties: (1) perimeter, surface area, and volume of the bounding box/sphere enclosing the geometric graph; (2) distance from the centroid or centre of mass; and (3) dihedral angles.*

**Practical Implications.** Proposition 10, together with Propositions 1 and 4, highlight critical theoretical limitations of $\mathfrak{G}$-invariant GNNs. Our results suggest that $\mathfrak{G}$-equivariant GNNs should be preferred when working with large geometric graphs such as macromolecules with thousands of nodes, where message passing is restricted to local radial neighbourhoods around each node.

Motivated by these limitations, two straightforward approaches to improving $\mathfrak{G}$-invariant GNNs may be: (1) pre-computing non-local geometric properties as input features, *e.g.* models such as GemNet [21] and GearNet [16] already use two-hop dihedral angles. And (2) working with fully connected geometric graphs, as Proposition 7 suggests that $\mathfrak{G}$-equivariant and $\mathfrak{G}$-invariant GNNs can be made equally powerful when performing all-to-all message passing. This is supported by the empirical success of recent $\mathfrak{G}$-invariant 'Graph Transformers' [22, 48] for small molecules with tens of nodes, where working with full graphs is tractable.

### C.2 Role of Body Order: Distinguishing $\mathfrak{G}$-Orbits

At each iteration of GWL and IGWL, the I-HASH function assigns a $\mathfrak{G}$-invariant colouring to distinct geometric neighbourhood patterns. I-HASH is an idealised $\mathfrak{G}$-orbit injective function which is not necessarily continous. In geometric GNNs, this corresponds to scalarising local geometric information when updating the scalar features; examples are shown in equation 9 and equation 10. We can analyse the construction of the I-HASH function and the scalarisation step in geometric GNNs via the $k$-body variations IGWL$_{(k)}$, described in Section 2.

Firstly, we formalise the relationship between the injectivity of I-HASH$_{(k)}$ and the maximum cardinality of local neighbourhoods in a given dataset.

**Proposition 11.** I-HASH$_{(m)}$ *is $\mathfrak{G}$-orbit injective for* $m = max(\{|\mathcal{N}_i| \mid i \in \mathcal{V}\})$, *the maximum cardinality of all local neighbourhoods $\mathcal{N}_i$ in a given dataset.*

**Practical Implications.** While building provably injective I-HASH$_{(k)}$ functions may require intractably high $k$, the hierarchy of IGWL$_{(k)}$ tests enable us to study the expressive power of practical $\mathfrak{G}$-invariant aggregators used in current geometric GNN layers, *e.g.* SchNet [13], E-GNN [11], and TFN [8] use distances, while DimeNet [15] uses distances and angles. Notably, MACE [20] constructs a *complete* basis of scalars up to arbitrary body order $k$ via Atomic Cluster Expansion [38], which can be $\mathfrak{G}$-orbit injective if the conditions in Proposition 11 are met. We can state the following about the IGWL$_{(k)}$ hierarchy and the corresponding GNNs.

**Proposition 12.** *IGWL$_{(k)}$ is at least as powerful as IGWL$_{(k-1)}$. For $k \leq 5$, IGWL$_{(k)}$ is strictly more powerful than IGWL$_{(k-1)}$.*

Finally, we show that IGWL$_{(2)}$ is equivalent to WL when all the pairwise distances between the nodes are the same. A similar observation was recently made by [49].

**Proposition 13.** *Let $\mathcal{G}_1 = (\boldsymbol{A}_1, \boldsymbol{S}_1, \vec{\boldsymbol{X}}_1)$ and $\mathcal{G}_2 = (\boldsymbol{A}_2, \boldsymbol{S}_2, \vec{\boldsymbol{X}}_2)$ be two geometric graphs with the property that all edges have equal length. Then, IGWL$_{(2)}$ distinguishes the two graphs if and only if WL can distinguish the attributed graphs $(\boldsymbol{A}_1, \boldsymbol{S}_1)$ and $(\boldsymbol{A}_1, \boldsymbol{S}_1)$.*

This equivalence points to limitations of distance-based $\mathfrak{G}$-invariant models like SchNet [13]. These models suffer from all well-known failure cases of WL, *e.g.* they cannot distinguish two equilateral triangles from the regular hexagon [15].

### C.3 Synthetic Experiment on Propagating Geometric Information

**$k$-chain geometric graphs.** GWL is an abstract theoretical tool capable of perfectly aggregating and propogating $\mathfrak{G}$-equivariant geometric information, which implies that the test can be run for any number of iterations without loss of information. In geometric GNNs, $\mathfrak{G}$-equivariant information is propogated via summing features from multiple layers in fixed dimensional spaces, which may lead to distortion or loss of information from distant nodes. To study the practical implications of depth in propagating geometric information, we consider $k$-chain geometric graphs which generalise the examples from [18]. Each pair of $k$-chains consists of $k + 2$ nodes with $k$ nodes arranged in a line and differentiated by the orientation of the 2 end points. Thus, $k$-chain graphs are $(\lfloor \frac{k}{2} \rfloor + 1)$-hop distinguishable, and $(\lfloor \frac{k}{2} \rfloor + 1)$ GWL iterations are theoretically sufficient to distinguish them.

**Setup and Hyperparameters.** We experiment with the following models: (1) SchNet [13] and DimeNet [15] as representative $\mathfrak{G}$-invariant GNNs; (2) E-GNN [11] and GVP-GNN [10] as representative $\mathfrak{G}$-equivariant GNNs which use cartesian vectors; and (3) TFN [8] and MACE [20] to study higher order $\mathfrak{G}$-equivariant GNNs using spherical tensors. For SchNet and DimeNet, we use the implementation from PyTorch Geometric [50]. For E-GNN, GVP-GNN, and MACE, we adapt implementations from the respective authors. Our TFN implementation is based on e3nn [46]. We set scalar feature channels to 128 for SchNet, DimeNet, and E-GNN. We set scalar/vector/tensor

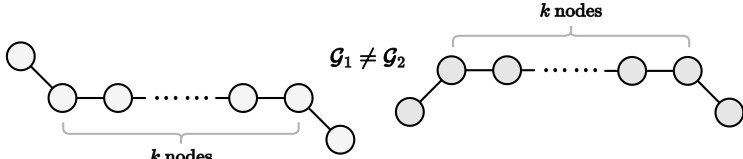

| | $(k = 4$-chains) | | **Number of layers** | | | |
|---|---|---|---|---|---|---|
| | **GNN Layer** | $\lfloor \frac{k}{2} \rfloor$ | $\lfloor \frac{k}{2} \rfloor + 1 = 3$ | $\lfloor \frac{k}{2} \rfloor + 2$ | $\lfloor \frac{k}{2} \rfloor + 3$ | $\lfloor \frac{k}{2} \rfloor + 4$ |
| | IGWL | 50% | 50% | 50% | 50% | 50% |
| Inv. | SchNet | $50.0 \pm 0.00$ | $50.0 \pm 0.00$ | $50.0 \pm 0.00$ | $50.0 \pm 0.00$ | $50.0 \pm 0.00$ |
| | DimeNet | $50.0 \pm 0.00$ | $50.0 \pm 0.00$ | $50.0 \pm 0.00$ | $50.0 \pm 0.00$ | $50.0 \pm 0.00$ |
| | GWL | 50% | 100% | 100% | 100% | 100% |
| | E-GNN | $50.0 \pm 0.0$ | $50.0 \pm 0.0$ | $50.0 \pm 0.0$ | $50.0 \pm 0.0$ | $100.0 \pm 0.0$ |
| Equiv. | GVP-GNN | $50.0 \pm 0.0$ | $100.0 \pm 0.0$ | $100.0 \pm 0.0$ | $100.0 \pm 0.0$ | $100.0 \pm 0.0$ |
| | TFN | $50.0 \pm 0.0$ | $50.0 \pm 0.0$ | $50.0 \pm 0.0$ | $80.0 \pm 24.5$ | $85.0 \pm 22.9$ |
| | MACE | $50.0 \pm 0.0$ | $90.0 \pm 20.0$ | $90.0 \pm 20.0$ | $95.0 \pm 15.0$ | $95.0 \pm 15.0$ |

**Table 1:** *$k$-chain geometric graphs.* $k$-chains are $(\lfloor \frac{k}{2} \rfloor + 1)$-hop distinguishable and $(\lfloor \frac{k}{2} \rfloor + 1)$ GWL iterations are theoretically sufficient to distinguish them. We train geometric GNNs with an increasing number of layers to distinguish $k = 4$-chains. **$\mathfrak{G}$-equivariant GNNs may require more iterations that prescribed by GWL, pointing to preliminary evidence of oversmoothing and oversquashing when geometric information is propagated across multiple layers using fixed dimensional feature spaces.** IGWL and $\mathfrak{G}$-invariant GNNs are unable to distinguish $k$-chains for any $k \geq 2$ and $\mathfrak{G} = O(3)$. Anomalous results are marked in red and expected results in green.

feature channels to 64 for GVP-GNN, TFN, MACE. TFN and MACE use order $L = 2$ tensors by default. MACE uses local body order 4 by default. We train all models for 100 epochs using the Adam optimiser, with an initial learning rate $1e - 4$, which we reduce by a factor of 0.9 and patience of 25 epochs when the performance plateaus. All results are averaged across 10 random seeds.

**Results.** In Table 1, we train $\mathfrak{G}$-equivariant and $\mathfrak{G}$-invariant GNNs with an increasing number of layers to distinguish $k$-chains. Despite the supposed simplicity of the task, especially for small chain lengths, we find that popular $\mathfrak{G}$-equivariant GNNs such as E-GNN and TFN may require more iterations that prescribed by GWL. Notably, as the length of the chain gets larger than $k = 4$, all $\mathfrak{G}$-equivariant GNNs tended to lose performance and required more than $(\lfloor \frac{k}{2} \rfloor + 1)$ iterations to solve the task. IGWL and $\mathfrak{G}$-invariant GNNs are unable to distinguish $k$-chains.

This synthetic experiment supplements our theoretical results and points to preliminary evidence of the *oversmoothing* and *oversquashing* phenomenon [51–53] for geometric GNNs. These issues are most evident for E-GNN, which uses a single vector feature to aggregate and propogate geometric information. This may have implications in modelling macromolecules where long-range interactions often play important roles. Studying both these issues are exciting avenues for future work towards building provably powerful, *practical* geometric GNNs.

# D  Proofs

## D.1  Proofs for What GWL and IGWL can Distinguish

The following results are a consequence of the construction of GWL as well as the definitions of $k$-hop distinct and $k$-hop identical geometric graphs. Note that $k$-hop distinct geometric graphs are also $(k + 1)$-hop distinct. Similarly, $k$-hop identical geometric graphs are also $(k - 1)$-hop identical, but not necessarily $(k + 1)$-hop distinct.

Given two distinct neighbourhoods $\mathcal{N}_1$ and $\mathcal{N}_2$, the $\mathfrak{G}$-orbits of the corresponding geometric multisets $\boldsymbol{g}_1$ and $\boldsymbol{g}_2$ are mutually exclusive, *i.e.* $\mathcal{O}_{\mathfrak{G}}(\boldsymbol{g}_1) \cap \mathcal{O}_{\mathfrak{G}}(\boldsymbol{g}_2) \equiv \emptyset$. By the properties of I-HASH this implies $c_1 \neq c_2$. Conversely, if $\mathcal{N}_1$ and $\mathcal{N}_2$ were identical up to group actions, their $\mathfrak{G}$-orbits would overlap, *i.e.* $\boldsymbol{g}_1 = \mathfrak{g}\, \boldsymbol{g}_2$ for some $\mathfrak{g} \in \mathfrak{G}$ and $\mathcal{O}_{\mathfrak{G}}(\boldsymbol{g}_1) = \mathcal{O}_{\mathfrak{G}}(\boldsymbol{g}_2) \Rightarrow c_1 = c_2$.

**Proposition 1.** *GWL can distinguish any $k$-hop distinct geometric graphs $\mathcal{G}_1$ and $\mathcal{G}_2$ where the underlying attributed graphs are isomorphic, and $k$ iterations are sufficient.*

*Proof of Proposition 1.* The $k$-th iteration of GWL identifies the $\mathfrak{G}$-orbit of the $k$-hop subgraph $\mathcal{N}_i^{(k)}$ at each node $i$ via the geometric multiset $\boldsymbol{g}_i^{(k)}$. $\mathcal{G}_1$ and $\mathcal{G}_2$ being $k$-hop distinct implies that there exists some bijection $b$ and some node $i \in \mathcal{V}_1, b(i) \in \mathcal{V}_2$ such that the corresponding $k$-hop subgraphs $\mathcal{N}_i^{(k)}$ and $\mathcal{N}_{b(i)}^{(k)}$ are distinct. Thus, the $\mathfrak{G}$-orbits of the corresponding geometric multisets $\boldsymbol{g}_i^{(k)}$ and $\boldsymbol{g}_{b(i)}^{(k)}$ are mutually exclusive, *i.e.* $\mathcal{O}_{\mathfrak{G}}(\boldsymbol{g}_i^{(k)}) \cap \mathcal{O}_{\mathfrak{G}}(\boldsymbol{g}_{b(i)}^{(k)}) \equiv \emptyset \Rightarrow c_i^{(k)} \neq c_{b(i)}^{(k)}$. Thus, $k$ iterations of GWL are sufficient to distinguish $\mathcal{G}_1$ and $\mathcal{G}_2$. $\square$

**Proposition 2.** *Up to $k$ iterations, GWL cannot distinguish any $k$-hop identical geometric graphs $\mathcal{G}_1$ and $\mathcal{G}_2$ where the underlying attributed graphs are isomorphic.*

*Proof of Proposition 2.* The $k$-th iteration of GWL identifies the $\mathfrak{G}$-orbit of the $k$-hop subgraph $\mathcal{N}_i^{(k)}$ at each node $i$ via the geometric multiset $\boldsymbol{g}_i^{(k)}$. $\mathcal{G}_1$ and $\mathcal{G}_2$ being $k$-hop identical implies that for all bijections $b$ and all nodes $i \in \mathcal{V}_1, b(i) \in \mathcal{V}_2$, the corresponding $k$-hop subgraphs $\mathcal{N}_i^{(k)}$ and $\mathcal{N}_{b(i)}^{(k)}$ are identical up to group actions. Thus, the $\mathfrak{G}$-orbits of the corresponding geometric multisets $\boldsymbol{g}_i^{(k)}$ and $\boldsymbol{g}_{b(i)}^{(k)}$ overlap, *i.e.* $\mathcal{O}_{\mathfrak{G}}(\boldsymbol{g}_i^{(k)}) = \mathcal{O}_{\mathfrak{G}}(\boldsymbol{g}_{b(i)}^{(k)}) \Rightarrow c_i^{(k)} = c_{b(i)}^{(k)}$. Thus, up to $k$ iterations of GWL cannot distinguish $\mathcal{G}_1$ and $\mathcal{G}_2$. $\square$

**Proposition 3.** *IGWL can distinguish any $1$-hop distinct geometric graphs $\mathcal{G}_1$ and $\mathcal{G}_2$ where the underlying attributed graphs are isomorphic, and $1$ iteration is sufficient.*

*Proof of Proposition 3.* Each iteration of IGWL identifies the $\mathfrak{G}$-orbit of the $1$-hop local neighbourhood $\mathcal{N}_i^{(k=1)}$ at each node $i$. $\mathcal{G}_1$ and $\mathcal{G}_2$ being $1$-hop distinct implies that there exists some bijection

$b$ and some node $i \in \mathcal{V}_1, b(i) \in \mathcal{V}_2$ such that the corresponding 1-hop local neighbourhoods $\mathcal{N}_i^{(1)}$ and $\mathcal{N}_{b(i)}^{(1)}$ are distinct. Thus, the $\mathfrak{G}$-orbits of the corresponding geometric multisets $\boldsymbol{g}_i^{(1)}$ and $\boldsymbol{g}_{b(i)}^{(1)}$ are mutually exclusive, *i.e.* $\mathcal{O}_{\mathfrak{G}}(\boldsymbol{g}_i^{(1)}) \cap \mathcal{O}_{\mathfrak{G}}(\boldsymbol{g}_{b(i)}^{(1)}) \equiv \emptyset \;\Rightarrow\; c_i^{(1)} \neq c_{b(i)}^{(1)}$. Thus, 1 iteration of IGWL is sufficient to distinguish $\mathcal{G}_1$ and $\mathcal{G}_2$. □

**Proposition 4.** *Any number of iterations of IGWL cannot distinguish any 1-hop identical geometric graphs $\mathcal{G}_1$ and $\mathcal{G}_2$ where the underlying attributed graphs are isomorphic.*

*Proof of Proposition 4.* Each iteration of IGWL identifies the $\mathfrak{G}$-orbit of the 1-hop local neighbourhood $\mathcal{N}_i^{(k=1)}$ at each node $i$, but cannot identify $\mathfrak{G}$-orbits beyond 1-hop by the construction of IGWL as no geometric information is propagated. $\mathcal{G}_1$ and $\mathcal{G}_2$ being 1-hop identical implies that for all bijections $b$ and all nodes $i \in \mathcal{V}_1, b(i) \in \mathcal{V}_2$, the corresponding 1-hop local neighbourhoods $\mathcal{N}_i^{(k)}$ and $\mathcal{N}_{b(i)}^{(k)}$ are identical up to group actions. Thus, the $\mathfrak{G}$-orbits of the corresponding geometric multisets $\boldsymbol{g}_i^{(1)}$ and $\boldsymbol{g}_{b(i)}^{(1)}$ overlap, *i.e.* $\mathcal{O}_{\mathfrak{G}}(\boldsymbol{g}_i^{(1)}) = \mathcal{O}_{\mathfrak{G}}(\boldsymbol{g}_{b(i)}^{(1)}) \;\Rightarrow\; c_i^{(k)} = c_{b(i)}^{(k)}$. Thus, any number of IGWL iterations cannot distinguish $\mathcal{G}_1$ and $\mathcal{G}_2$. □

**Proposition 5.** *Assuming geometric graphs are constructed from point clouds using radial cutoffs, GWL can distinguish any geometric graphs $\mathcal{G}_1$ and $\mathcal{G}_2$ where the underlying attributed graphs are non-isomorphic. At most $k_{Max}$ iterations are sufficient, where $k_{Max}$ is the maximum graph diameter among $\mathcal{G}_1$ and $\mathcal{G}_2$.*

*Proof of Proposition 5.* We assume that a geometric graph $\mathcal{G} = (\boldsymbol{A}, \boldsymbol{S}, \vec{\boldsymbol{V}}, \vec{\boldsymbol{X}})$ is constructed from a point cloud $(\boldsymbol{S}, \vec{\boldsymbol{V}}, \vec{\boldsymbol{X}})$ using a predetermined radial cutoff $r$. Thus, the adjacency matrix is defined as $a_{ij} = 1$ if $\|\vec{\boldsymbol{x}}_i - \vec{\boldsymbol{x}}_j\|_2 \leq r$, or 0 otherwise, for all $a_{ij} \in \boldsymbol{A}$. Such construction procedures are conventional for geometric graphs in biochemistry and material science.

Given geometric graphs $\mathcal{G}_1$ and $\mathcal{G}_2$ where the underlying attributed graphs are non-isomorphic, identify $k_{\text{Max}}$ the maximum of the graph diameters of $\mathcal{G}_1$ and $\mathcal{G}_2$, and chose any arbitrary nodes $i \in \mathcal{V}_1, j \in \mathcal{V}_2$. We can define the $k_{\text{Max}}$-hop subgraphs $\mathcal{N}_i^{(k_{\text{Max}})}$ and $\mathcal{N}_j^{(k_{\text{Max}})}$ at $i$ and $j$, respectively. Thus, $\mathcal{N}_i^{(k_{\text{Max}})} = \mathcal{V}_1$ for all $i \in \mathcal{V}_1$, and $\mathcal{N}_j^{(k_{\text{Max}})} = \mathcal{V}_2$ for all $j \in \mathcal{V}_2$. Due to the assumed construction procedure of geometric graphs, $\mathcal{N}_i^{(k_{\text{Max}})}$ and $\mathcal{N}_j^{(k_{\text{Max}})}$ must be distinct. Otherwise, if $\mathcal{N}_i^{(k_{\text{Max}})}$ and $\mathcal{N}_j^{(k_{\text{Max}})}$ were identical up to group actions, the sets $(\boldsymbol{S}_1, \vec{\boldsymbol{V}}_1, \vec{\boldsymbol{X}}_1)$ and $(\boldsymbol{S}_2, \vec{\boldsymbol{V}}_2, \vec{\boldsymbol{X}}_2)$ would have yielded isomorphic graphs.

The $k_{\text{Max}}$-th iteration of GWL identifies the $\mathfrak{G}$-orbit of the $k_{\text{Max}}$-hop subgraph $\mathcal{N}_i^{(k_{\text{Max}})}$ at each node $i$ via the geometric multiset $\boldsymbol{g}_i^{(k_{\text{Max}})}$. As $\mathcal{N}_i^{(k_{\text{Max}})}$ and $\mathcal{N}_j^{(k_{\text{Max}})}$ are distinct for any arbitrary nodes $i \in \mathcal{V}_1, j \in \mathcal{V}_2$, the $\mathfrak{G}$-orbits of the corresponding geometric multisets $\boldsymbol{g}_i^{(k_{\text{Max}})}$ and $\boldsymbol{g}_j^{(k_{\text{Max}})}$ are mutually exclusive, *i.e.* $\mathcal{O}_{\mathfrak{G}}(\boldsymbol{g}_i^{(k_{\text{Max}})}) \cap \mathcal{O}_{\mathfrak{G}}(\boldsymbol{g}_j^{(k_{\text{Max}})}) \equiv \emptyset \;\Rightarrow\; c_i^{(k_{\text{Max}})} \neq c_j^{(k_{\text{Max}})}$. Thus, $k_{\text{Max}}$ iterations of GWL are sufficient to distinguish $\mathcal{G}_1$ and $\mathcal{G}_2$. □

**Theorem 6.** *GWL is strictly more powerful than IGWL.*

*Proof of Theorem 6.* Firstly, we can show that the GWL class contains IGWL if GWL can learn the identity when updating $\boldsymbol{g}_i$ for all $i \in \mathcal{V}$, *i.e.* $\boldsymbol{g}_i^{(t)} = \boldsymbol{g}_i^{(t-1)} = \boldsymbol{g}_i^{(0)} \equiv (\boldsymbol{s}_i, \vec{\boldsymbol{v}}_i)$. Thus, GWL is at least as powerful as IGWL, which does not update $\boldsymbol{g}_i$.

Secondly, to show that GWL is strictly more powerful than IGWL, it suffices to show that there exist a pair of geometric graphs that can be distinguished by GWL but not by IGWL. We may consider any $k$-hop distinct geometric graphs for $k > 1$, where the underlying attributed graphs are isomorphic. Proposition 1 states that GWL can distinguish any such graphs, while Proposition 4 states that IGWL cannot distinguish them. An example is the pair of graphs in Figures 1 and 2. □

**Proposition 7.** *IGWL has the same expressive power as GWL for fully connected geometric graphs.*

*Proof of Proposition 7.* We will prove by contradiction. Assume that there exist a pair of fully connected geometric graphs $\mathcal{G}_1$ and $\mathcal{G}_2$ which GWL can distinguish, but IGWL cannot.

If the underlying attributed graphs of $\mathcal{G}_1$ and $\mathcal{G}_2$ are isomorphic, by Proposition 1 and Proposition 4, $\mathcal{G}_1$ and $\mathcal{G}_2$ are 1-hop identical but $k$-hop distinct for some $k > 1$. For all bijections $b$ and all nodes $i \in \mathcal{V}_1, b(i) \in \mathcal{V}_2$, the local neighbourhoods $\mathcal{N}_i^{(1)}$ and $\mathcal{N}_{b(i)}^{(1)}$ are identical up to group actions, and $\mathcal{O}_{\mathfrak{G}}(\boldsymbol{g}_i^{(1)}) = \mathcal{O}_{\mathfrak{G}}(\boldsymbol{g}_{b(i)}^{(1)}) \Rightarrow c_i^{(1)} = c_{b(i)}^{(1)}$. Additionally, there exists some bijection $b$ and some nodes $i \in \mathcal{V}_1, b(i) \in \mathcal{V}_2$ such that the $k$-hop subgraphs $\mathcal{N}_i^{(k)}$ and $\mathcal{N}_{b(i)}^{(k)}$ are distinct, and $\mathcal{O}_{\mathfrak{G}}(\boldsymbol{g}_i^{(k)}) \cap \mathcal{O}_{\mathfrak{G}}(\boldsymbol{g}_{b(i)}^{(k)}) \equiv \emptyset \Rightarrow c_i^{(k)} \neq c_{b(i)}^{(k)}$. However, as $\mathcal{G}_1$ and $\mathcal{G}_2$ are fully connected, for any $k$, $\mathcal{N}_i^{(1)} = \mathcal{N}_i^{(k)}$ and $\mathcal{N}_{b(i)}^{(1)} = \mathcal{N}_{b(i)}^{(k)}$ are identical up to group actions. Thus, $\mathcal{O}_{\mathfrak{G}}(\boldsymbol{g}_i^{(1)}) = \mathcal{O}_{\mathfrak{G}}(\boldsymbol{g}_i^{(k)}) = \mathcal{O}_{\mathfrak{G}}(\boldsymbol{g}_{b(i)}^{(1)}) = \mathcal{O}_{\mathfrak{G}}(\boldsymbol{g}_{b(i)}^{(k)}) \Rightarrow c_i^{(1)} = c_i^{(k)} = c_{b(i)}^{(k)} = c_{b(i)}^{(k)}$. This is a contradiction.

If $\mathcal{G}_1$ and $\mathcal{G}_2$ are non-isomorphic and fully connected, for any arbitrary $i \in \mathcal{V}_1, j \in \mathcal{V}_2$ and any $k$-hop neighbourhood, we know that $\mathcal{N}_i^{(1)} = \mathcal{N}_i^{(k)}$ and $\mathcal{N}_j^{(1)} = \mathcal{N}_j^{(k)}$. Thus, a single iteration of GWL and IGWL identify the same $\mathfrak{G}$-orbits and assign the same node colours, *i.e.* $\mathcal{O}_{\mathfrak{G}}(\boldsymbol{g}_i^{(1)}) = \mathcal{O}_{\mathfrak{G}}(\boldsymbol{g}_i^{(k)}) \Rightarrow c_i^{(1)} = c_i^{(k)}$ and $\mathcal{O}_{\mathfrak{G}}(\boldsymbol{g}_j^{(1)}) = \mathcal{O}_{\mathfrak{G}}(\boldsymbol{g}_j^{(k)}) \Rightarrow c_j^{(1)} = c_j^{(k)}$. This is a contradiction. $\qquad\square$

## D.2 Proofs for equivalence between GWL and Geometric GNNs

Our proofs adapt the techniques used in Xu et al. [25], Morris et al. [26] for connecting WL with GNNs. Note that we omit including the relative position vectors $\vec{x}_{ij}$ in GWL and geometric GNN updates for brevity, as relative positions vectors can be merged into the vector features.

**Theorem 8.** *Any pair of geometric graphs distinguishable by a $\mathfrak{G}$-equivariant GNN is also distinguishable by GWL.*

***Proof of Theorem 8.*** Consider two geometric graphs $\mathcal{G}$ and $\mathcal{H}$. The theorem implies that if the GNN graph-level readout outputs $f(\mathcal{G}) \neq f(\mathcal{H})$, then the GWL test will always determine $\mathcal{G}$ and $\mathcal{H}$ to be non-isomorphic, *i.e.* $\mathcal{G} \neq \mathcal{H}$.

We will prove by contradiction. Suppose after $T$ iterations, a GNN graph-level readout outputs $f(\mathcal{G}) \neq f(\mathcal{H})$, but the GWL test cannot decide $\mathcal{G}$ and $\mathcal{H}$ are non-isomorphic, *i.e.* $\mathcal{G}$ and $\mathcal{H}$ always have the same collection of node colours for iterations 0 to $T$. Thus, for iteration $t$ and $t+1$ for any $t = 0 \ldots T-1$, $\mathcal{G}$ and $\mathcal{H}$ have the same collection of node colours $\{c_i^{(t)}\}$ as well as the same collection of neighbourhood geometric multisets $\left\{ (c_i^{(t)}, \boldsymbol{g}_i^{(t)}) , \ \{\!\!\{ (c_j^{(t)}, \boldsymbol{g}_j^{(t)}) \mid j \in \mathcal{N}_i \}\!\!\} \right\}$ up to group actions. Otherwise, the GWL test would have produced different node colours at iteration $t+1$ for $\mathcal{G}$ and $\mathcal{H}$ as different geometric multisets get unique new colours.

We will show that on the same graph for nodes $i$ and $k$, if $(c_i^{(t)}, \boldsymbol{g}_i^{(t)}) = (c_k^{(t)}, \mathfrak{g} \cdot \boldsymbol{g}_k^{(t)})$, we always have GNN features $(\boldsymbol{s}_i^{(t)}, \vec{\boldsymbol{v}}_i^{(t)}) = (\boldsymbol{s}_k^{(t)}, \boldsymbol{Q}_{\mathfrak{g}} \vec{\boldsymbol{v}}_k^{(t)})$ for any iteration $t$. This holds for $t = 0$ because GWL and the GNN start with the same initialisation. Suppose this holds for iteration $t$. At iteration $t+1$, if for any $i$ and $k$, $(c_i^{(t+1)}, \boldsymbol{g}_i^{(t+1)}) = (c_k^{(t+1)}, \mathfrak{g} \cdot \boldsymbol{g}_k^{(t+1)})$, then:

$$\left\{ (c_i^{(t)}, \boldsymbol{g}_i^{(t)}) , \ \{\!\!\{ (c_j^{(t)}, \boldsymbol{g}_j^{(t)}) \mid j \in \mathcal{N}_i \}\!\!\} \right\} = \left\{ (c_k^{(t)}, \mathfrak{g} \cdot \boldsymbol{g}_k^{(t)}) , \ \{\!\!\{ (c_j^{(t)}, \mathfrak{g} \cdot \boldsymbol{g}_j^{(t)}) \mid j \in \mathcal{N}_k \}\!\!\} \right\} \quad (18)$$

By our assumption on iteration $t$,

$$\left\{ (\boldsymbol{s}_i^{(t)}, \vec{\boldsymbol{v}}_i^{(t)}) , \ \{\!\!\{ (\boldsymbol{s}_j^{(t)}, \vec{\boldsymbol{v}}_j^{(t)}) \mid j \in \mathcal{N}_i \}\!\!\} \right\} = \left\{ (\boldsymbol{s}_k^{(t)}, \boldsymbol{Q}_{\mathfrak{g}} \vec{\boldsymbol{v}}_k^{(t)}) , \ \{\!\!\{ (\boldsymbol{s}_j^{(t)}, \boldsymbol{Q}_{\mathfrak{g}} \vec{\boldsymbol{v}}_j^{(t)}) \mid j \in \mathcal{N}_k \}\!\!\} \right\} \quad (19)$$

As the same aggregate and update operations are applied at each node within the GNN, the same inputs, *i.e.* neighbourhood features, are mapped to the same output. Thus, $(\boldsymbol{s}_i^{(t+1)}, \vec{\boldsymbol{v}}_i^{(t+1)}) = (\boldsymbol{s}_k^{(t+1)}, \boldsymbol{Q}_{\mathfrak{g}} \vec{\boldsymbol{v}}_k^{(t+1)})$. By induction, if $(c_i^{(t)}, \boldsymbol{g}_i^{(t)}) = (c_k^{(t)}, \mathfrak{g} \cdot \boldsymbol{g}_k^{(t)})$, we always have GNN node features $(\boldsymbol{s}_i^{(t)}, \vec{\boldsymbol{v}}_i^{(t)}) = (\boldsymbol{s}_k^{(t)}, \boldsymbol{Q}_{\mathfrak{g}} \vec{\boldsymbol{v}}_k^{(t)})$ for any iteration $t$. This creates valid mappings $\phi_s, \phi_v$ such that $\boldsymbol{s}_i^{(t)} = \phi_s(c_i^{(t)})$ and $\vec{\boldsymbol{v}}_i^{(t)} = \phi_v(c_i^{(t)}, \boldsymbol{g}_i^{(t)})$ for any $i \in \mathcal{V}$.

Thus, if $\mathcal{G}$ and $\mathcal{H}$ have the same collection of node colours and geometric multisets, then $\mathcal{G}$ and $\mathcal{H}$ also have the same collection of GNN neighbourhood features

$$\left\{(\boldsymbol{s}_i^{(t)}, \vec{\boldsymbol{v}}_i^{(t)}), \ \{\!\!\{(\boldsymbol{s}_j^{(t)}, \vec{\boldsymbol{v}}_j^{(t)}) \mid j \in \mathcal{N}_i\}\!\!\}\right\} = \left\{(\phi_s(c_i^{(t)}), \phi_v(c_i^{(t)}, \boldsymbol{g}_i^{(t)})), \ \{\!\!\{(\phi_s(c_j^{(t)}), \phi_v(c_i^{(t)}, \boldsymbol{g}_i^{(t)})) \mid j \in \mathcal{N}_i\}\!\!\}\right\}$$

Thus, the GNN will output the same collection of node scalar features $\{\boldsymbol{s}_i^{(T)}\}$ for $\mathcal{G}$ and $\mathcal{H}$ and the permutation-invariant graph-level readout will output $f(\mathcal{G}) = f(\mathcal{H})$. This is a contradiction. $\qquad\square$

Similarly, $\mathfrak{G}$-invariant GNNs can be at most as powerful as IGWL.

**Theorem 14.** *Any pair of geometric graphs distinguishable by a $\mathfrak{G}$-invariant GNN is also distinguishable by IGWL.*

*Proof.* The proof follows similarly to the proof for Theorem 8. $\qquad\square$

**Proposition 9.** *$\mathfrak{G}$-equivariant GNNs have the same expressive power as GWL if the following conditions hold: (1) The aggregation* AGG *is an injective, $\mathfrak{G}$-equivariant multiset function. (2) The scalar part of the update* $\mathrm{UPD}_s$ *is a $\mathfrak{G}$-orbit injective, $\mathfrak{G}$-invariant multiset function. (3) The vector part of the update* $\mathrm{UPD}_v$ *is an injective, $\mathfrak{G}$-equivariant multiset function. (4) The graph-level readout $f$ is an injective multiset function.*

***Proof of Theorem 9.*** Consider a GNN where the conditions hold. We will show that, with a sufficient number of iterations $t$, the output of this GNN is equivalent to GWL, *i.e.* $\boldsymbol{s}^{(t)} \equiv c^{(t)}$.

Let $\mathcal{G}$ and $\mathcal{H}$ be any geometric graphs which the GWL test decides as non-isomorphic at iteration $T$. Because the graph-level readout function is injective, *i.e.* it maps distinct multiset of node scalar features into unique embeddings, it suffices to show that the GNN's neighbourhood aggregation process, with sufficient iterations, embeds $\mathcal{G}$ and $\mathcal{H}$ into different multisets of node features.

For this proof, we replace $\mathfrak{G}$-orbit injective functions with injective functions over the equivalence class generated by the actions of $\mathfrak{G}$. Thus, all elements belonging to the same $\mathfrak{G}$-orbit will first be mapped to the same representative of the equivalence class, denoted by the square brackets $[\ldots]$, followed by an injective map. The result is $\mathfrak{G}$-orbit injective.

Let us assume the GNN updates node scalar and vector features as:

$$\boldsymbol{s}_i^{(t)} = \mathrm{UPD}_s\left(\left[(\boldsymbol{s}_i^{(t-1)}, \vec{\boldsymbol{v}}_i^{(t-1)}), \ \mathrm{AGG}\left(\{\!\!\{(\boldsymbol{s}_i^{(t-1)}, \boldsymbol{s}_j^{(t-1)}, \vec{\boldsymbol{v}}_i^{(t-1)}, \vec{\boldsymbol{v}}_j^{(t-1)}) \mid j \in \mathcal{N}_i\}\!\!\}\right)\right]\right) \quad (20)$$

$$\vec{\boldsymbol{v}}_i^{(t)} = \mathrm{UPD}_v\left((\boldsymbol{s}_i^{(t-1)}, \vec{\boldsymbol{v}}_i^{(t-1)}), \ \mathrm{AGG}\left(\{\!\!\{(\boldsymbol{s}_i^{(t-1)}, \boldsymbol{s}_j^{(t-1)}, \vec{\boldsymbol{v}}_i^{(t-1)}, \vec{\boldsymbol{v}}_j^{(t-1)}) \mid j \in \mathcal{N}_i\}\!\!\}\right)\right) \quad (21)$$

with the aggregation function AGG being $\mathfrak{G}$-equivariant and injective, the scalar update function $\mathrm{UPD}_s$ being $\mathfrak{G}$-invariant and injective, and the vector update function $\mathrm{UPD}_v$ being $\mathfrak{G}$-equivariant and injective.

The GWL test updates the node colour $c_i^{(t)}$ and geometric multiset $\boldsymbol{g}_i^{(t)}$ as:

$$c_i^{(t)} = h_s\left(\left[(c_i^{(t-1)}, \boldsymbol{g}_i^{(t-1)}), \ \{\!\!\{(c_j^{(t-1)}, \boldsymbol{g}_j^{(t-1)}) \mid j \in \mathcal{N}_i\}\!\!\}\right]\right), \quad (22)$$

$$\boldsymbol{g}_i^{(t)} = h_v\left((c_i^{(t-1)}, \boldsymbol{g}_i^{(t-1)}), \ \{\!\!\{(c_j^{(t-1)}, \boldsymbol{g}_j^{(t-1)}) \mid j \in \mathcal{N}_i\}\!\!\}\right), \quad (23)$$

where $h_s$ is a $\mathfrak{G}$-invariant and injective map, and $h_v$ is a $\mathfrak{G}$-equivariant and injective operation (e.g. in equation 2, expanding the geometric multiset by copying).

We will show by induction that at any iteration $t$, there always exist injective functions $\varphi_s$ and $\varphi_v$ such that $\boldsymbol{s}_i^{(t)} = \varphi_s(c_i^{(t)})$ and $\vec{\boldsymbol{v}}_i^{(t)} = \varphi_v(c_i^{(t)}, \boldsymbol{g}_i^{(t)})$. This holds for $t = 0$ because the initial node features are the same for GWL and GNN, $c_i^{(0)} \equiv \boldsymbol{s}_i^{(0)}$ and $\boldsymbol{g}_i^{(0)} \equiv (\boldsymbol{s}_i^{(0)}, \vec{\boldsymbol{v}}_i^{(0)})$ for all $i \in \mathcal{V}(\mathcal{G}), \mathcal{V}(\mathcal{H})$. Suppose this holds for iteration $t$. At iteration $t+1$, substituting $\boldsymbol{s}_i^{(t)}$ with $\varphi_s(c_i^{(t)})$, and $\vec{\boldsymbol{v}}_i^{(t)}$ with $\varphi_v(c_i^{(t)}, \boldsymbol{g}_i^{(t)})$ gives us

$$\boldsymbol{s}_i^{(t+1)} = \mathrm{UPD}_s\left(\left[(\varphi_s(c_i^{(t)}), \varphi_v(c_i^{(t)}, \boldsymbol{g}_i^{(t)})), \ \mathrm{AGG}\left(\{\!\!\{(\varphi_s(c_i^{(t)}), \varphi_s(c_j^{(t)}), \varphi_v(c_i^{(t)}, \boldsymbol{g}_i^{(t)}), \varphi_v(c_j^{(t)}, \boldsymbol{g}_j^{(t)})) \mid j \in \mathcal{N}_i\}\!\!\}\right)\right]\right)$$

$$\vec{\boldsymbol{v}}_i^{(t+1)} = \mathrm{UPD}_v\left((\varphi_s(c_i^{(t)}), \varphi_v(c_i^{(t)}, \boldsymbol{g}_i^{(t)})), \ \mathrm{AGG}\left(\{\!\!\{(\varphi_s(c_i^{(t)}), \varphi_s(c_j^{(t)}), \varphi_v(c_i^{(t)}, \boldsymbol{g}_i^{(t)}), \varphi_v(c_j^{(t)}, \boldsymbol{g}_j^{(t)})) \mid j \in \mathcal{N}_i\}\!\!\}\right)\right)$$

The composition of multiple injective functions is injective. Therefore, there exist some injective functions $g_s$ and $g_v$ such that:

$$s_i^{(t+1)} = g_s \left( \left[ (c_i^{(t)}, \boldsymbol{g}_i^{(t)}), \ \{\!\{ (c_j^{(t)}, \boldsymbol{g}_j^{(t)}) \mid j \in \mathcal{N}_i \}\!\} \right] \right), \tag{24}$$

$$\vec{\boldsymbol{v}}_i^{(t+1)} = g_v \left( (c_i^{(t)}, \boldsymbol{g}_i^{(t)}), \ \{\!\{ (c_j^{(t)}, \boldsymbol{g}_j^{(t)}) \mid j \in \mathcal{N}_i \}\!\} \right), \tag{25}$$

We can then consider:

$$s_i^{(t+1)} = g_s \circ h_s^{-1} \, h_s \left( \left[ (c_i^{(t)}, \boldsymbol{g}_i^{(t)}), \ \{\!\{ (c_j^{(t)}, \boldsymbol{g}_j^{(t)}) \mid j \in \mathcal{N}_i \}\!\} \right] \right), \tag{26}$$

$$\vec{\boldsymbol{v}}_i^{(t+1)} = g_v \circ h_v^{-1} \, h_v \left( (c_i^{(t)}, \boldsymbol{g}_i^{(t)}), \ \{\!\{ (c_j^{(t)}, \boldsymbol{g}_j^{(t)}) \mid j \in \mathcal{N}_i \}\!\} \right), \tag{27}$$

Then, we can denote $\varphi_s = g_s \circ h_s^{-1}$ and $\varphi_v = g_v \circ h_v^{-1}$ as injective functions because the composition of injective functions is injective. Hence, for any iteration $t + 1$, there exist injective functions $\varphi_s$ and $\varphi_v$ such that $s_i^{(t+1)} = \varphi_s \left( c_i^{(t+1)} \right)$ and $\vec{\boldsymbol{v}}_i^{(t+1)} = \varphi_v \left( c_i^{(t+1)}, \boldsymbol{g}_i^{(t+1)} \right)$.

At the $T$-th iteration, the GWL test decides that $\mathcal{G}$ and $\mathcal{H}$ are non-isomorphic, which means the multisets of node colours $\{c_i^{(T)}\}$ are different for $\mathcal{G}$ and $\mathcal{H}$. The GNN's node scalar features $\{s_i^{(T)}\} = \{\varphi_s(c_i^{(T)})\}$ must also be different for $\mathcal{G}$ and $\mathcal{H}$ because of the injectivity of $\varphi_s$.

$\square$

A weaker set of conditions is sufficient for a $\mathfrak{G}$-invariant GNN to be at least as expressive as IGWL.

**Proposition 15.** *$\mathfrak{G}$-invariant GNNs have the same expressive power as IGWL if the following conditions hold: (1) The aggregation $\psi$ and update $\phi$ are $\mathfrak{G}$-orbit injective, $\mathfrak{G}$-invariant multiset functions. (2) The graph-level readout $f$ is an injective multiset function.*

*Proof.* The proof follows similarly to the proof for Theorem 9. $\square$

### D.3 Geometric GNN Design Space Proofs

**Proposition 10.** *IGWL and $\mathfrak{G}$-invariant GNNs cannot decide several geometric graph properties: (1) perimeter, surface area, and volume of the bounding box/sphere enclosing the geometric graph; (2) distance from the centroid or centre of mass; and (3) dihedral angles.*

*Proof of Proposition 10.* Following Garg et al. [47], we say that a class of models *decides* a geometric graph property if there exists a model belonging to this class such that for any two geometric graphs that differ in the property, the model is able to distinguish the two geometric graphs.

In Figure 4 we provide an example of two geometric graphs that demonstrate the proposition. $\mathcal{G}_1$ and $\mathcal{G}_2$ differ in the following geometric graph properties:

- Perimeter, surface area, and volume of the bounding box enclosing the geometric graph[3]: (32 units, 40 units$^2$, 16 units$^3$) vs. (28 units, 24 units$^2$, 8 units$^3$).

- Multiset of distances from the centroid or centre of mass: $\{0.00, 1.00, 1.00, 2.45, 2.45\}$ vs. $\{0.40, 1.08, 1.08, 2.32, 2.32\}$.

- Dihedral angles: $\angle(ljkm) = \frac{(\vec{\boldsymbol{x}}_{jk} \times \vec{\boldsymbol{x}}_{lj}) \cdot (\vec{\boldsymbol{x}}_{jk} \times \vec{\boldsymbol{x}}_{mk})}{|\vec{\boldsymbol{x}}_{jk} \times \vec{\boldsymbol{x}}_{lj}| |\vec{\boldsymbol{x}}_{jk} \times \vec{\boldsymbol{x}}_{mk}|}$ are clearly different for the two graphs.

However, according to Proposition 4 and Theorem 14, both IGWL and $\mathfrak{G}$-invariant GNNs cannot distinguish these two geometric graphs, and therefore, cannot decide all these properties.

We can also show this by constructing geometric computation trees for any number of IGWL or $\mathfrak{G}$-invariant GNN iterations, as illustrated in Figure 3. Recall that a computation tree $\mathcal{T}_i^{(t)}$ represents the maximum information contained in GWL/IGWL colours or GNN features for node $i$ at iteration $t$ by an 'unrolling' of the message passing procedure. Geometric computation trees are constructed

---

[3]The same result applies for the bounding sphere, not shown in the figure.

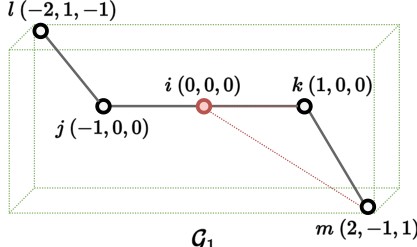 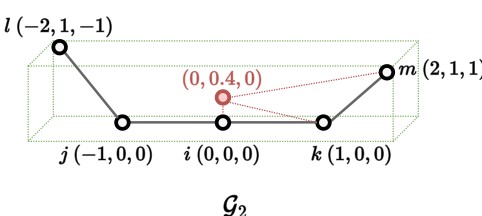

**Figure 4:** Two geometric graphs for which IGWL and $\mathfrak{G}$-invariant GNNs cannot distinguish their perimeter, surface area, volume of the bounding box/sphere, distance from the centroid, and dihedral angles. The centroid is denoted by a red point and distances from it are denoted by dotted red lines. The bounding box enclosing the geometric graph is denoted by the dotted green lines.

recursively: $\mathcal{T}_i^{(0)} = (\boldsymbol{s}_i, \vec{\boldsymbol{v}}_i)$ for all $i \in \mathcal{V}$. For $t > 0$, we start with a root node $(\boldsymbol{s}_i, \vec{\boldsymbol{v}}_i)$ and add a child subtree $\mathcal{T}_j^{(t-1)}$ for all $j \in \mathcal{N}_i$ along with the relative position $\vec{\boldsymbol{x}}_{ij}$ along the edge. To obtain the root node's embedding or colour, both scalar and geometric information is propagated from the leaves up to the root. Thus, if two nodes have identical geometric computation trees, they will be mapped to the same node embedding or colour.

Critically, geometric orientation information cannot flow from one level to another in the computation trees for IGWL and $\mathfrak{G}$-invariant GNNs, as they only update scalar information. In the recursive construction procedure, we must insert a connector node $(\boldsymbol{s}_j, \vec{\boldsymbol{v}}_j)$ before adding the child subtree $\mathcal{T}_j^{(t-1)}$ for all $j \in \mathcal{N}_i$ and prevent geometric information propagation between them.

Following the construction procedure for the geometric graphs in Figure 4, we observe that the IGWL computation trees of any pair of isomorphic nodes are identical, as all 1-hop neighbourhoods are computationally identical. Therefore, the set of node colours or node scalar features will also be identical, which implies that $\mathcal{G}_1$ and $\mathcal{G}_2$ cannot be distinguished. $\qquad\square$

**Proposition 11.** I-HASH$_{(m)}$ *is $\mathfrak{G}$-orbit injective for* $m = max(\{|\mathcal{N}_i| \mid i \in \mathcal{V}\})$*, the maximum cardinality of all local neighbourhoods* $\mathcal{N}_i$ *in a given dataset.*

*Proof of Proposition 11.* As $m$ is the maximum cardinality of all local neighbourhoods $\mathcal{N}_i$ under consideration, any distinct neighbourhoods $\mathcal{N}_1$ and $\mathcal{N}_2$ must have distinct multisets of $m$-body scalars. As I-HASH$_{(m)}$ computes scalars involving up to $m$ nodes, it will be able to distinguish any such $\mathcal{N}_1$ and $\mathcal{N}_2$. Thus, I-HASH$_{(m)}$ is $\mathfrak{G}$-orbit injective. $\qquad\square$

**Proposition 12.** *IGWL$_{(k)}$ is at least as powerful as IGWL$_{(k-1)}$. For $k \leq 5$, IGWL$_{(k)}$ is strictly more powerful than IGWL$_{(k-1)}$.*

*Proof of Proposition 12.* By construction, I-HASH$_{(k)}$ computes $\mathfrak{G}$-invariant scalars from all possible tuples of up to $k$ nodes formed by the elements of a neighbourhood and the central node. Thus, the I-HASH$_{(k)}$ class contains I-HASH$_{(k-1)}$, and I-HASH$_{(k)}$ is at least as powerful as I-HASH$_{(k-1)}$. Thus, the corresponding test IGWL$_{(k)}$ is at least as powerful as IGWL$_{(k-1)}$.

Secondly, to show that IGWL$_{(k)}$ is strictly more powerful than IGWL$_{(k-1)}$ for $k \leq 5$, it suffices to show that there exist a pair of geometric neighbourhoods that can be distinguished by IGWL$_{(k)}$ but not by IGWL$_{(k-1)}$:

- For $k = 3$ and $\mathfrak{G} = O(3)$ or $SO(3)$, for the local neighbourhood from Figure 1 in [18], two configurations with different angles between the neighbouring nodes can be distinguished by IGWL$_{(3)}$ but not by IGWL$_{(2)}$.

- For $k = 4$ and $\mathfrak{G} = O(3)$ or $SO(3)$, the pair of local neighbourhoods from Figure 1 in [35] can be distinguished by IGWL$_{(4)}$ but not by IGWL$_{(3)}$.

- For $k = 5$ and $\mathfrak{G} = O(3)$, the pair of local neighbourhoods from Figure 2(e) in [35] can be distinguished by IGWL$_{(5)}$ but not by IGWL$_{(4)}$.

- For $k = 5$ and $\mathfrak{G} = SO(3)$, the pair of local neighbourhoods from Figure 2(f) in [35] can be distinguished by $\text{IGWL}_{(5)}$ but not by $\text{IGWL}_{(4)}$.

$\square$

**Proposition 13.** *Let $\mathcal{G}_1 = (\boldsymbol{A}_1, \boldsymbol{S}_1, \vec{\boldsymbol{X}}_1)$ and $\mathcal{G}_2 = (\boldsymbol{A}_2, \boldsymbol{S}_2, \vec{\boldsymbol{X}}_2)$ be two geometric graphs with the property that all edges have equal length. Then, $\text{IGWL}_{(2)}$ distinguishes the two graphs if and only if WL can distinguish the attributed graphs $(\boldsymbol{A}_1, \boldsymbol{S}_1)$ and $(\boldsymbol{A}_1, \boldsymbol{S}_1)$.*

*Proof of Proposition 13.* Let $c$ and $k$ the colours produced by $\text{IGWL}_{(2)}$ and WL, respectively, and let $i$ and $j$ be two nodes belonging to any two graphs like in the statement of the result. We prove the statement inductively.

Clearly, $c_i^{(0)} = k_i^{(0)}$ for all nodes $i$ and $c_i^{(0)} = c_j^{(0)}$ if and only if $k_i^{(0)} = k_j^{(0)}$. Now, assume that the statement holds for iteration $t$. That is $c_i^{(t)} = c_j^{(t)}$ if and only if $k_i^{(t)} = k_j^{(t)}$ holds for all $i$. Note that $c_i^{(t+1)} = c_j^{(t+1)}$ if and only if $c_i^{(t)} = c_j^{(t)}$ and $\{\!\!\{(c_p^{(t)}, \|\vec{\boldsymbol{x}}_{ip}\|) \mid p \in \mathcal{N}_i\}\!\!\} = \{\!\!\{(c_p^{(t)}, \|\vec{\boldsymbol{x}}_{jp}\|) \mid p \in \mathcal{N}_j\}\!\!\}$, since the norm of the relative vectors is the only injective invariant that $\text{IGWL}_{(2)}$ can compute (up to a scaling). Since all the norms are equal, by the induction hypothesis, this is equivalent to $k_i^{(t)} = k_j^{(t)}$ and $\{\!\!\{k_p^{(t)} \mid p \in \mathcal{N}_i\}\!\!\} = \{\!\!\{k^{(t)} \mid p \in \mathcal{N}_j\}\!\!\}$. Therefore, this is equivalent to $k_i^{(t+1)} = k_j^{(t+1)}$ $\square$

