# OpenReview forum: "On the Expressive Power of Geometric Graph Neural Networks"
_logconference.io/LOG/2022/Conference — LoG 2022 Poster_

### Official Review · Reviewer_eqmz · 2022-10-13

**Overall Score:** 6
**Confidence:** 4

**Review:**

## Overview:
The authors consider the problem of graph isomorphism for graphs embedded in some metric space (geometric graphs), where two graphs are considered isomorphic if the edge-preserving bijection also leads to node features that are equivalent up to global symmetries.
They introduce a novel modification of the Weisfeiler-Leman (WL) test which they deem Geometric Weisfeiler-Leman (GWL) test.
The main technical contribution of the paper is a comparison between the power of GWL and invariant/equivariant GNNs, showing that equivariant GNNs can be at most as powerful as GWL.

## Strengths:
The authors consider a problem that is important for a large variety of applications relevant to GNNs. They present a natural extension of the WL test and situate invariant/equivariant GNNs in the context of the proposed test.

## Weaknesses:
### Clarity/writing style and motivation
Unfortunately, I believe that the paper is poorly written in its current form. In particular, the lack of introduction and overall terse discussion in my opinion do not do an adequate job at motivating the work and situating it withing the context of the existing literature. While I understand that the page limit is extremely restrictive, I strongly believe that the authors can strike a better balance between formal definitions/preliminaries and a high-level motivation/description of the problem.

In particular, on multiple occasions the authors make an informal assumption that the graph is "constructed under realistic assumptions" which is then interpreted as "constructed from point clouds using radial cutoffs". While this is indeed a natural class of graphs to consider for certain applications (e.g. neural force fields for molecular dynamics (MD) simulations), it is by far not an exhaustive list of "realistic assumptions" in general. Thus I believe that the authors can strengthen the paper by emphasizing results like Proposition 7 and making the MD application a more explicit motivation.

### Context
The proposed GWL test is a natural extension to the WL test, and the authors cite relevant works relating GNNs and WL tests in the absence of geometric considerations. However, I believe that the authors need to do a more thorough comparison with existing works on geometric graph isomorphism/matching, e.g. [1].

### Practicality
As the authors suggest, it seems unclear whether current GNN architectures can satisfy the conditions outlined in this paper for GNNs to be as powerful as GWL. I would have appreciated a longer discussion around "we suspect that higher-order tensors are required for building functions with the desired properties" (line 154).

## Statements I do not understand
- Line 63-64: "Since a $\mathfrak{G}$-invariant function cannot be injective by construction, this aggregator must be $\mathfrak{G}$-equivariant." Why do we need the aggregator function to be equivariant? If our definition of isomorphism "doesn't care" about global group action, can we not get away with an invariant aggregator?
- Eq. (2): Why do we need to keep track of $c_i$ inside of $g_i$? I.e. why can I not set $g_i^{(0)}=(v_i, \emptyset)$ and $g_i^{(t)}=(g_i^{(t-1)}, \\{g_j^{(t-1)}, x_{ij} | j\in \mathcal{N}_i\\})$? Additionally, using this definition, $g^{(t)}$ seems to explode quite fast in size as $t$ grows? Do we want the LHS of the tuple to also be a multiset?

## Additional comments/questions:
- Line 16: the notation $\mathfrak{G}=SO(d)/O(d)$ is very confusing because it appears to refer to the (nonsensical) quotient $SO(d)/O(d)$ instead of the intended "the group $\mathfrak{G}$ can be either $SO(d)$ or $O(d)$. The issue persists throughout the paper.
- Eq. (3) and (4) and other places: is the notation $\\{\\{\cdot\\}\\}$ used to indicate a multiset?
- Line 54-55: What do you mean by "we assume that all geometric vectors and scalar features come from a countable subset of $\mathbb{R}^d$ and $\mathbb{R}^f$"? This seems very restrictive -- why is it needed?
- Line 62-63: In "to avoid any loss of information, the local aggregator must be permutation-invariant and injective", how does lack of permutation-invariance lead to "loss of information"?
- The overall approach and notation in this paper bears strong resemblance to [2] (which does not consider geometry). I believe it would be nice to make that inspiration more explicit in the paper.
- Is there a separation between invariant and equivariant GNNs in the absence of tensor/vector features $V$, i.e. if all features are scalar?

## Minor comments:
- Line 27: it would be nice to define semidirect product (or at lest spell out with words) before using $\rtimes$.
- Line 83: typo ("is only updates")

## Recommendation:
In my opinion, the stated weakness outweigh the strengths of the manuscript in its current form and hence I recommend rejection. I am happy to update my score when the comments/questions are addressed.

[1] J. J. McAuley, T. de Campos and T. S. Caetano, "Unified graph matching in Euclidean spaces," 2010 IEEE Computer Society Conference on Computer Vision and Pattern Recognition, 2010, pp. 1871-1878, doi: 10.1109/CVPR.2010.5539859.
[2] Morris, Christopher, et al. “Weisfeiler and Leman Go Neural: Higher-Order Graph Neural Networks.” Proceedings of the AAAI Conference on Artificial Intelligence, vol. 33, July 2019, pp. 4602–09. Crossref, https://doi.org/10.1609/aaai.v33i01.33014602.

--------------

# Post-rebuttal
I would like to thank the authors for thoroughly addressing my comments and questions. The updated manuscript is substantially more readable. Thus, I am satisfied regarding R4.1 and R4.2.

The discussion of novelty wrt [2] could be made stronger by highlighting the main challenges but overall I am satisfied with R4.3.

I am thoroughly impressed with how the authors have addressed R4.4 -- the discussion and synthetic experiment bring much needed clarity to the discussion from the previous version of the manuscript.

I am satisfied to the answers provided to questions/comments R4.5-R4.13.

Overall, I believe that the authors have significantly improved the paper and thus I update my score to recommend acceptance!

---

### Official Review · Reviewer_UN6P · 2022-10-15

**Overall Score:** 5
**Confidence:** 3

**Review:**

## Overview:

The author attempts to propose a geometric version of Weisfeiler-Leman graph isomorphism test (GWL) to handle disorder and rigid transfer in geometric graph analysis task. Some prior knowledge of Weisfeiler-Leman graph isomorphism test (WL) are used to characterize the expressive power of GNN. In fact, I don't have enough professional knowledge of WL. Therefore, I find a related reference [1] and try to understand the basic thought. It seems like the algorithm is used to recode the graph structure and reveal the essential relationships between different nodes. Based on the understanding, I regret to say that the article cannot be published in current version. The comments are as follow.

## Comments:

1) In Sec. 1, the author provides the definition of geometric graph that includes scalar features and tensor of vector features. However, there have no further explanations about the parameters. Does the two items have relationships? Does the scalar features are necessary? The author should provide an instance at least.
2) In contributions, the author said “We propose a geometric version of the Weisfeiler-Leman graph isomorphism test (GWL)”. For my understanding, the GWL adds some geometric features into the WL and make it more practical in geometric features-based GNN tasks with. It is just a simple extension to WL.
3) The author provides relevant evidences to illustrate the function of GWL for graph isomorphism test or graph-based feature analysis. Considering that the LoG still falls with the scope of engineering, the author should provide more engineering experiments, such as non-rigid shape analysis, 3D topology analysis, object recognition, knowledge graph, etc.

[1] Huang N T, Villar S. A Short Tutorial on The Weisfeiler-Lehman Test And Its Variants[C]//ICASSP 2021-2021 IEEE International Conference on Acoustics, Speech and Signal Processing (ICASSP). IEEE, 2021: 8533-8537.

---

## Reply to the rebuttal:

In fact, I'm still not satisfied with the arrangement of the article. It should be clear that the content in Appendix should be regarded as the additional information, including proof of detail and explanation of extreme cases. Therefore, I still stand by my opinion that the current version is not suitable for publication, especially for a conference with engineering background.

---

### Official Review · Reviewer_RpQs · 2022-10-18

**Overall Score:** 6
**Confidence:** 5

**Review:**

### **Summary**:

This paper introduces a theoretical framework to analyse Geometric GNNs, i.e., GNNs that are also invariant or equivariant to geometric transformations (in addition to isomorphism). In particular, the authors propose Geometric WL, akin to the classical WL algorithm that has been widely used to formalise the expressive power of GNNs. The authors show that Geometric WL can be used as a conceptual model to upper bound the expressive power of Geometric GNNs. Furthermore, they show the capabilities and limitations of GWL in distinguishing geometric graphs, illustrating the interplay between simple graph isomorphism and geometric graph isomorphism.

### **Strengths**:
Recently there has been a surge of papers proposing different neural networks with built-in equivariances simultaneously to continuous and combinatorial symmetries for applications in physics, chemistry etc. However, most of these papers, focus on the formulation of the methods in order to achieve equivariance, and in some cases in the computational complexity, but the expressive power is overlooked. Given the above, this paper has the following advantages:

- *Novelty*: It marks a first step (to the best of my knowledge), in formally analysing the expressive power of Geometric GNNs. The Geometric WL framework closely follows the GNN analyses and is a simple and unifying way to provide theoretical results.
- *Importance & Insights*: The results regarding the classes of graphs that can/cannot be distinguished are insightful and are a good starting point for deeper studies. Furthermore, deriving expressive Geometric GNNs from first principles deserves more attention from the community and can provide researchers with an extra tool to compare their approaches, in addition to the empirical results.

### **Weaknesses**:

*Clarity*: Unfortunately, the presentation of the paper is poor and makes it hard to follow. The flow is not very good, there are several notation issues and some concepts are inadequately presented. Given that it is submitted as an extended abstract, this might be a factor of smaller importance when it comes to the decision, but I strongly encourage the authors to proofread, simplify and improve the presentation of their work in an updated version. Some suggestions:
- Provide examples of Geometric GNNs presented in the literature and show how they can be formulated under this framework. This is important to show the relevance of this analysis. Which of these Geometric GNNs are G-equivariant and which are G-invariant?
- Fix potential notation issues and inconsistencies. Also, many concepts are inadequately defined or defined in the appendix. For example:
    - Please define G ⋊ T(d). Also, the notation SO(d)/O(d) might be confusing (I assume it means “either”),
     - Definition 11: First off, perhaps this should have been included in the main text, and second: isn't it better to frame it as: “there exists $\mathfrak{g}$ and $\vec{\mathbf{t}}$ such that Eq (8) holds”?
     - Eq (4): why isn’t there a HASH function here as well? Why is a HASH function used in the proof of Theorem 2 only? In the same proof, $h_s$ and $h_u$ are undefined but I assume that they stand for HASH functions.
     - Theorem 2: The functions $\phi_s$, $\phi_u$ and $f$ are undefined and need to be inferred from the context. Also, (minor) in the proof, half of the equations include a square bracket [], which I assume is a typo.

*Technical depth*: Although the framework of Geometric WL is reasonable and useful, the actual technical innovation is limited. In particular, Theorems 1 and 2 are direct adaptations from Xu et al., ICLR’20, while Proposition 3-Corollary 9, can be easily derived from the definitions. I believe there is space for much deeper studies. For example, the assumptions of Proposition 7 are probably too restrictive (it should be possible to arrive at a similar result for arbitrary graph constructions, assuming that the geometric features allow vertex disambiguation).

### **Other comments**:

- It will be beneficial to compare specific geometric GNN architectures w.r.t. their expressivity since this is a question of interest to the community.
- I am a bit puzzled about the vector features V. Why do they necessarily have the same dimensionality (and live in the same space) as the coordinates X? Why does the reflection/rotation group act on the vertex features but the translation group doesn’t?
It will be helpful to provide an example.
- Why is translation equivariance/invariance overlooked? Is the GWL framework adequate for maximal expressivity in that respect?

### **Overall**:

The present manuscript has several clarity issues and lacks sufficient technical innovation. Currently, I am leaning towards rejection, but given that it is an extended abstract, I might consider increasing my score in case the authors can provide certain improvements in their manuscript (along the lines mentioned above). In any case, the overall research direction of formally/theoretically analysing Geometric GNNs is interesting so I would encourage the authors to delve deeper into the theoretical questions that arise.

### **Post-Rebuttal**:

Most of my concerns have been addressed and the presentation of the manuscript has improved considerably. I increased my score to recommend acceptance and I am looking forward to the extension of this work that the authors are currently working on.

---

### Official Review · Reviewer_FvRA · 2022-10-21

**Overall Score:** 8
**Confidence:** 4

**Review:**

This paper proposes the concept of geometric Weisfeiler-Leman (GWL) extending the 2D graph isomorphism test algorithm to
distinguish 3D graphs with different geometric structures.
Based on this concept, the author analyzes and compares the expressiveness of invariant and equivariant Graph Neural Networks.
At last, several conditions are proposed to achieve GWL-level GNNs.

Pro:
The proposed GWL is an interesting idea, and the expressiveness power of geometric graphs plays a vital role in 3D graph modeling.
This paper provides solid theory support for the GWL test,  and proposes one kind of characterization of 3D GNN with GWL-level.

Cons:
Although the theory is solid, there is no experiment to verify the proposed GWL-level GNNs.

Comments:
The complexity analysis is also important to analyze the proposed GWL algorithm.

### Post-Rebuttal:
I appreciate the author's hard work to improve this paper. All my concerns are solved. And I raise my score to support the acceptance.

---

### Meta-Review · Area_Chair_dmkv · 2022-11-16

**Confidence:** 4
**Recommendation:** Accept

**Meta Review:**

Although there were some concerns regarding the "extended abstract" format, the reviewers were satisfied with the improvement brought after the first round of review. We recommend acceptance.

---

### Decision · Program_Chairs · 2022-11-23

Accept (Poster)